EMBO
reports

# S1P-S1PR1 signaling impairs CD8+ T cell metabolism and effector function in tumors

Debashree Basak[1,2], Puspendu Ghosh [1], Anupam Gautam [3,4], Ishita Sarkar[1,2], Arpita Bhoumik [1], Soham Chowdhury [1,2], Shaun Mahanti [1,2], Anwesha Mandal [1,2], Rajeswari Chakraborty[1,2], Anwesha Kar [1], Snehanshu Chowdhury [1], Krishna Kumar [5], Shubhrajit Barman [2,5], Senthil Kumar Ganesan [5], Saikat Chakrabarti[5], Sandip Paul [6] & Shilpak Chatterjee [1,2 ✉]

## Abstract

Sphingosine-1-phosphate receptor 1 (S1PR1) signaling has been linked to the regulation of immunosuppressive cell populations within the tumor microenvironment (TME); however, its role in shaping anti-tumor CD8+ T cell responses remains poorly defined. Herein, we demonstrate that intratumoral CD8+ T cells express S1PR1, with expression predominantly enriched in the terminally exhausted subset. Transcriptomic profiling, combined with pharmacological inhibition and genetic knockdown, reveals that S1PR1-S1P signaling activates the PERK (protein kinase R (PKR)-like endoplasmic reticulum kinase)-CHOP (C/EBP homologous protein) axis of the endoplasmic reticulum stress response. CHOP, in turn, upregulates transcription of *Map3k13* and *Map3k15*, triggering downstream MAPK signaling and culminating in activation of p38MAPK. Activation of this pathway impairs CD8+ T cell metabolism and effector function while increasing apoptotic susceptibility. This ultimately limits the persistence and accumulation of functional CD8+ T cells within the TME, thereby compromising their responsiveness to anti-PD-1 therapy. Targeting the S1PR1-S1P axis or its downstream effectors offers a promising strategy to improve cancer immunotherapy outcomes.

**Keywords** S1P-S1PR1; CD8+ T Cells; ER Stress
**Subject Categories** Cancer; Immunology; Signal Transduction

## Introduction

As tumors evolve, they accumulate genetic alterations, such as somatic mutations and chromosomal rearrangements, which make them increasingly distinct from normal cells (Zhang et al, 2024). This molecular divergence often renders them immunologically visible, triggering a tumor-specific CD8+ T cell (TST) response aimed at recognizing and eliminating malignant cells (Dunn et al, 2004; Shankaran et al, 2001). However, despite the presence of TSTs, tumors continue to progress, suggesting that these T cells become functionally impaired in the tumor microenvironment (TME) (Schreiber et al, 2011; Thommen and Schumacher, 2018). Breakthrough discoveries unraveling the molecular basis of T cell dysfunction in tumors have led to immunotherapeutic strategies designed to reinvigorate TST function (Leach et al, 1996; Nishimura et al, 1999; Pardoll, 2012). Specifically, immune checkpoint blockade (ICB) therapy targeting PD-1 or CTLA-4, alone or in combination, has demonstrated remarkable clinical success across various malignancies (Sharma et al, 2023). However, despite these advances, ICB benefits only a fraction of patients; the majority either fail to respond or experience tumor relapse. This underscores an urgent need to elucidate the cellular mechanisms driving T cell dysfunction in cancer. A deeper understanding of these processes will be crucial for devising therapeutic strategies to enhance T cell responsiveness and improve immunotherapeutic outcomes.

Sphingosine-1-phosphate (S1P) is a bioactive lipid that regulates numerous physiological processes, including angiogenesis, hematopoiesis, carcinogenesis, tumor metastasis, and immune modulation (Mendelson et al, 2014). Synthesized by sphingosine kinases (SphK1 and SphK2), S1P exerts its effects by binding to G protein-coupled receptors (S1PR1-S1PR5), initiating diverse signaling cascades (Proia and Hla, 2015; Spiegel and Milstien, 2007). Among these receptors, S1PR1 is predominant in T cells and plays a key role in T cell dynamics. Specifically, S1PR1 governs the egress of naive and activated T cells from lymphoid organs into circulation, ensuring their availability for immune surveillance (Benechet et al, 2016; Pham et al, 2008; Thangada et al, 2010). In addition to its role in T cell egress, the S1P-S1PR1 axis is crucial in modulating the differentiation and effector functions of CD4+ T cell subsets (Garris et al, 2013; Huang et al, 2007; Liu et al, 2009; Liu et al, 2010). However, the precise regulatory mechanisms of this axis remain unclear, as contradictory findings have been reported.

[1]Division of Cancer Biology and Inflammatory Disorder, IICB-Translational Research Unit of Excellence; CSIR-Indian Institute of Chemical Biology, Kolkata, India. [2]Academy of Scientific and Innovative Research (AcSIR), Ghaziabad, India. [3]Institute for Bioinformatics and Medical Informatics, University of Tübingen, Tübingen, Germany. [4]International Max Planck Research School "From Molecules to Organisms", Max Planck Institute for Biology Tübingen, Max-Planck-Ring 5, Tübingen, Germany. [5]Division of Structural Biology and Bioinformatics, IICB-Translational Research Unit of Excellence; CSIR-Indian Institute of Chemical Biology, Kolkata, India. [6]Center for Health Science and Technology, JIS Institute of Advanced Studies and Research, JIS University, Kolkata, India. ✉E-mail: schatterjee@iicb.res.in

Recently, the S1P-S1PR1 signaling pathway has gained attention for its role in shaping the immune landscape of tumors. Notably, S1PR1-mediated activation of JAK2/STAT3 signaling has been shown to promote the accumulation of regulatory T cells (Tregs) in the tumor microenvironment (TME) (Liu et al, 2019; Priceman et al, 2014). Similarly, studies in bladder carcinoma patients reported elevated S1PR1 expression on Tregs, underscoring its role in enhancing Treg functionality (Liu et al, 2019). Beyond Tregs, S1PR1 signaling has been implicated in fostering the accumulation of myeloid-derived suppressor cells (MDSCs) in the TME, thereby facilitating tumor progression in gastric cancer (Zhou and Guo, 2018). While these findings highlight the pivotal role of the S1P-S1PR1 axis in regulating the migration and functionality of various immune subsets within the TME, its impact on CD8$^+$ T cell functionality remains largely unexplored.

In this study, we investigated the role of the S1P-S1PR1 axis in regulating CD8$^+$ T cell functionality in tumors, prompted by our observation that intratumoral CD8$^+$ T cells exhibit high S1PR1 surface expression. Strikingly, S1P-S1PR1 signaling not only impaired CD8$^+$ T cell function but also increased their susceptibility to cell death, critically limiting their ability to mount a durable anti-tumor response. Our findings reveal a previously unrecognized role of the S1P-S1PR1 axis in controlling intratumoral CD8$^+$ T cell persistence and suggest that targeting this pathway or its downstream signaling could enhance the therapeutic efficacy of CD8$^+$ T cells in tumors.

## Results

### S1PR1 is highly expressed on intratumoral CD8$^+$ T cells, and its activation dampens the function and metabolic fitness of CD8$^+$ T cells

To decipher the relevance of S1P-S1PR1 signaling axis, if any, in regulating the functional and phenotypic traits of intratumoral CD8$^+$ T cells, we first examined S1PR1 expression on CD8$^+$ T cells across various preclinical tumor models, including EL-4 lymphoma, B16-F10 melanoma, and the aggressive, immunotherapy-resistant YUMM1.7 melanoma model that harbors human-relevant mutations. Analysis of antigen-experienced (CD44$^+$) CD8$^+$ T cells revealed that S1PR1 expression was significantly elevated on intratumoral CD8$^+$ T cells compared to their splenic counterparts across all tumor types examined (Fig. 1A–C). Notably, this difference was more pronounced in CD8$^+$ T cells isolated from B16F10 (Fig. 1B) and YUMM1.7 (Fig. 1C) mouse melanomas, two highly aggressive, pre-clinical tumor models refractory to anti-PD1 therapy (Kar et al, 2024). Given the heterogeneous exhaustion states of CD8$^+$ T cells within the tumor microenvironment (TME) distinguished by the differential expression of PD1 and Tim3, we next investigated whether S1PR1 expression was enriched in specific subsets. Notably, terminally exhausted CD8$^+$ T cells (PD1$^+$Tim3$^+$) exhibited significantly higher surface levels of S1PR1 compared to progenitor exhausted cells (PD1$^-$Tim3$^-$) (Figs. 1D–F and EV1A–C). To determine whether this trend was preserved in vitro, we assessed chronically stimulated CD8$^+$ T cells. Consistent with our in vivo findings, both human and murine CD8$^+$ T cells subjected to chronic TCR stimulation, models mimicking distinct features of exhaustion (Fig. EV1D–M), exhibited elevated

S1PR1 surface expression relative to acutely stimulated cells (Fig. EV1N,O). Furthermore, given that STAT3 activation is known to sustain S1PR1 expression, we observed that intratumoral CD8$^+$ T cells having high S1PR1 expression displayed elevated levels of p-STAT3 (Fig. EV1P).

We next examined whether S1PR1 expression correlates with T cell activation, as it has been reported to increase following antigen encounter (Pham et al, 2008). To this end, we first checked the transcript levels of all five S1prs (S1pr1, S1pr2, S1pr3, S1pr4, and S1pr5) in CD8$^+$ T cells over a time course of in vitro activation to comprehensively map their expression dynamics. Interestingly, among all receptors, S1pr1 transcript levels showed a progressive, time-dependent increase upon activation (Fig. 1G). This observation was corroborated at the protein level by assessing S1PR1 expression on CD8$^+$ T cells at different time-points of their activation (Fig. 1H). Given the elevated S1PR1 expression on activated CD8$^+$ T cells, we next explored the functional consequences of S1PR1 signaling. To this end, we supplemented activated CD8$^+$ T cell cultures with exogenous S1P and evaluated effector cytokine production. Remarkably, S1P-treated cells exhibited significantly reduced production of IFN-γ, granzyme B (GZMB), and TNF-α compared to vehicle controls (Fig. 1I). Since proliferation is integral to T cell effector function, we also assessed the impact of S1P treatment on CD8$^+$ T cell expansion. S1P exposure led to a marked reduction in proliferation (Fig. 1J), consistent with decreased CD25 expression on activated CD8$^+$ T cells (Fig. EV1Q). Most notably, triggering the S1P-S1PR1 axis dramatically increased CD8$^+$ T cell susceptibility to death following TCR re-stimulation (Fig. 1K).

Metabolic regulation is closely tied to T cell effector function. To investigate how S1P-S1PR1 signaling influences the metabolic features of CD8$^+$ T cells, we first assessed glucose uptake using the fluorescent glucose analogue 2-NBDG. CD8$^+$ T cells activated in the presence of exogenous S1P showed significantly reduced 2-NBDG uptake (Fig. EV2A), suggesting that S1P impairs glycolytic capacity. Supporting this, extracellular acidification rate (ECAR) measurements, a direct measure of glycolysis, revealed a notable decline in glycolytic activity following S1P treatment (Fig. 1L–N). This metabolic impairment was further corroborated at the transcriptional level, where key glycolytic genes were downregulated in S1P-treated CD8$^+$ T cells (Fig. 1O). Beyond glycolysis, we also examined mitochondrial function, which is frequently compromised in dysfunctional T cells. Strikingly, mitochondrial morphology in S1P-treated CD8$^+$ T cells appeared disrupted, characterized by a diffuse, rounded structure, in contrast to the dense, elongated mitochondria observed in vehicle-treated controls (Fig. EV2B). We reasoned that this altered morphology might result from enhanced mitochondrial fission, as Drp1, a gene associated with fission, was significantly upregulated in S1P-treated CD8$^+$ T cells, whereas the expression of mitochondrial fusion genes (Opa1, Mfn1, and Mfn2) was downregulated (Fig. EV2C). Consistent with this, oxygen consumption rate (OCR), a readout of mitochondrial oxidative phosphorylation (OXPHOS), was significantly reduced in CD8$^+$ T cells exposed to S1P (Fig. 1P–R). Additionally, spare respiratory capacity (SRC), a marker of mitochondrial fitness and long-term T cell persistence, was markedly diminished (Fig. 1S). Finally, S1P-treated CD8$^+$ T cells exhibited a sharp decrease in ATP production (Fig. EV2D), further confirming mitochondrial dysfunction.

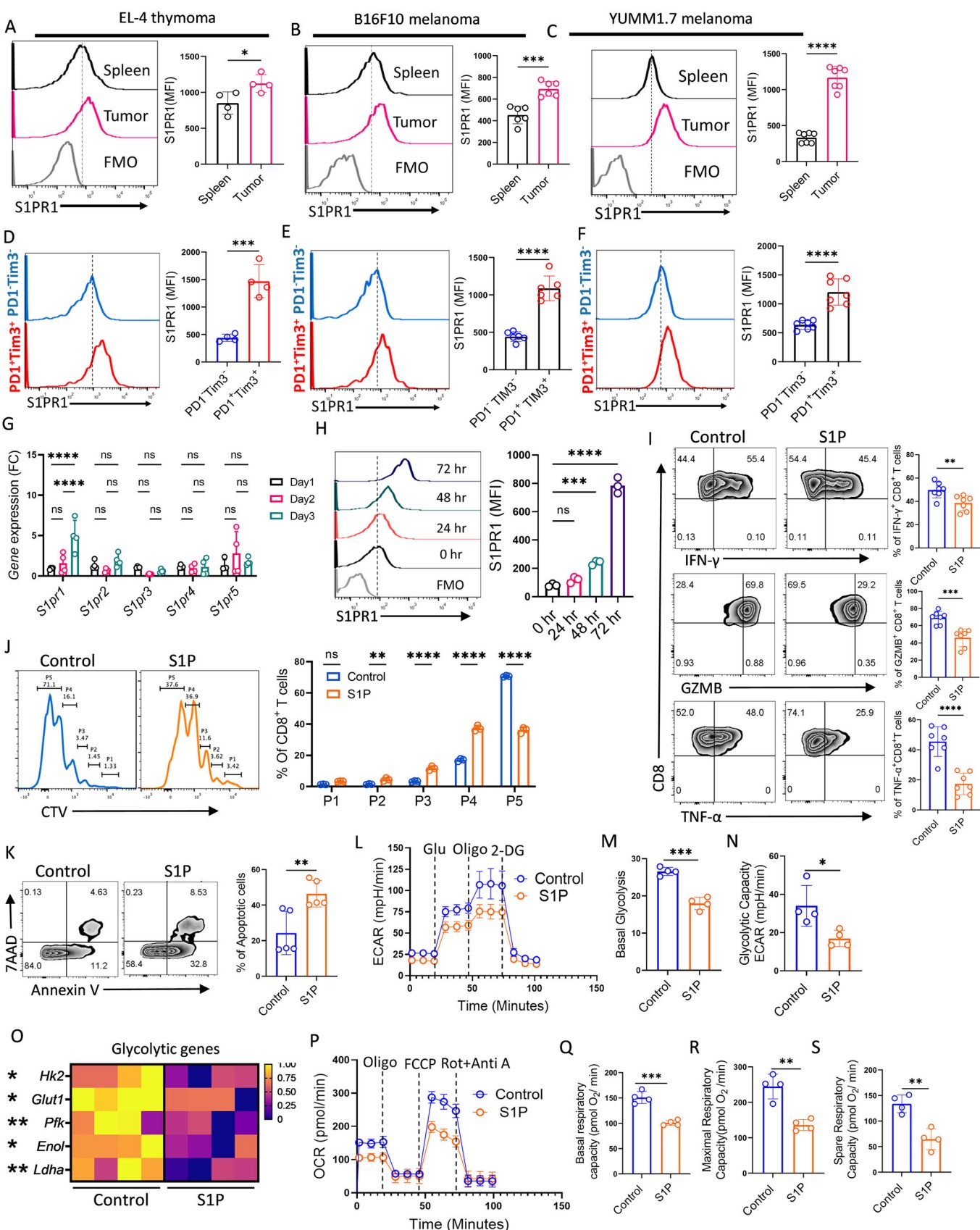

◄  **Figure 1.   Elevated S1PR1 expression on intratumoral CD8⁺ T cells compromise their functionality and anti-tumor efficacy.**

(A–C) Flow cytometric analysis of S1PR1 expression in tumor tissues and spleens of B6 mice bearing 15-day subcutaneously established tumors: (A) EL4 thymoma ($n = 4$), (B) B16 melanoma ($n = 6$), and (C) YUMM1.7 melanoma ($n = 7$). (D–F) Assessment of S1PR1 expression in terminally exhausted (PD1⁺Tim3⁺) and progenitor-like exhausted (PD1⁻Tim3⁻) CD8⁺ T cells isolated from (D) EL4 thymoma ($n = 4$, $P = 0.0005$), (E) B16 melanoma ($n = 6$), and (F) YUMM1.7 melanoma ($n = 7$). (G) Purified CD8⁺ T cells from wild-type B6 mice were activated in vitro for 3 days, and transcript levels of *S1pr1-S1pr5* were quantified by qPCR ($n = 4$). (H) Purified CD8⁺ T cells from wild-type B6 mice were activated in vitro for 3 days, and cell-surface S1PR1 expression was examined by flow cytometry ($n = 3$). (I) Flow cytometry-based intracellular cytokine analysis of in vitro-activated murine CD8⁺ T cells ($n = 7$). (J) Proliferative capacity of CTV-labeled mouse CD8⁺ T cells activated in the presence or absence of S1P, assessed using flow cytometry ($n = 3$). (K) Flow cytometry analysis of apoptosis in mouse CD8⁺ T cells activated with or without S1P, quantified by Annexin V and 7-AAD staining ($n = 5$). (L–N) Purified mouse CD8⁺ T cells activated in the presence or absence of S1P were subjected to metabolic flux analysis to determine: (L) extracellular acidification rate (ECAR) in response to glucose, oligomycin, and 2-deoxyglucose (2DG); (M) basal glycolytic rate following glucose addition ($n = 4$); and (N) glycolytic capacity ($n = 4$). (O) Heatmap illustrating transcript levels of key glycolysis-associated genes in S1P-treated versus vehicle-treated mouse CD8⁺ T cells ($n = 4$). (P–S) Oxygen consumption rate (OCR) of mouse CD8⁺ T cells activated in the presence or absence of S1P under basal conditions and after sequential addition of mitochondrial inhibitors (P), along with quantification of (Q) basal respiration ($n = 4$), (R) maximal respiratory capacity ($n = 4$), and (S) spare respiratory capacity ($n = 4$, $P = 0.0021$). *$P < 0.05$; **$P < 0.01$; ***$P < 0.005$; ****$P < 0.0001$; ns, nonsignificant ($P > 0.05$), the error bar represents the standard deviation (SD). $P$ values are derived from unpaired two-tailed Student's $t$ test (A–F, I, K, M, N, Q–S), one-way ANOVA (H), and two-way ANOVA test (G, J, O). Source data are available online for this figure.

Taken together, these findings demonstrate that S1PR1, which is highly expressed on exhausted CD8⁺ T cells, suppresses the effector function and profoundly impairs the metabolic fitness of CD8⁺ T cells upon activation by its ligand, S1P.

## Disruption of the S1P-S1PR1 signaling axis markedly improves the anti-tumor potential of CD8⁺ T cells

Given that intratumoral CD8⁺ T cells express S1PR1 and that activation of the S1P-S1PR1 signaling axis impairs their effector function and metabolic fitness, we next investigated whether disrupting this pathway could enhance their anti-tumor potential. Although S1PR1 is essential for naive T cell egress from the thymus and lymph nodes, as well as for peripheral trafficking of effector T cells, systemic inhibition of S1PR1 leads to lymphodepletion (Matloubian et al, 2004). To circumvent this, we aimed to selectively disrupt the S1P-S1PR1 axis in TME by locally depleting S1P levels.

Previous studies have shown that S1P is enriched in the TME and promotes tumor growth and metastasis (Schneider, 2020). While S1P is synthesized from sphingosine by two isozymes, sphingosine kinase 1 (SphK1) and sphingosine kinase 2 (SphK2) (Kohama et al, 1998; Liu et al, 2000), extracellular S1P is primarily generated by SphK1, which is frequently overexpressed in tumor cells (Wang et al, 2020). To locally reduce S1P in the TME, we used CRISPR-Cas9 to generate a SphK1 knockout in the YUMM1.7 melanoma cell line (SphK1^KO YUMM1.7) (Fig. EV3A). To test whether SphK1 loss affects tumorigenicity, we subcutaneously engrafted Rag1⁻/⁻ mice (lacking mature T and B cells) with either wild-type (WT) or SphK1^KO YUMM1.7 cells. Tumor growth kinetics were comparable in both groups, indicating that SphK1 loss did not impair tumor cell-intrinsic growth (Fig. 2A).

To evaluate the effect of SphK1 loss on anti-tumor immunity, we next implanted WT and SphK1^KO YUMM1.7 cells into immunocompetent C57BL/6 (B6) mice. Tumors lacking SphK1 exhibited significantly delayed growth, suggestive of enhanced anti-tumor immune activity (Fig. 2B). Supporting this, the frequency of antigen-experienced (CD44⁺) CD8⁺ T cells was significantly higher in SphK1^KO tumors (Fig. 2C). Further phenotypic and functional analysis revealed that intratumoral CD8⁺ T cells from SphK1^KO tumors displayed enhanced polyfunctionality (i.e., co-production of multiple effector cytokines) (Fig. 2D) and a less exhausted phenotype, with reduced expression of Tim3 compared to their WT counterparts (Fig. 2E).

To directly confirm the role of S1PR1 in regulating CD8⁺ T cell dysfunction, we locally inhibited S1PR1 signaling using W146, a pharmacological antagonist, administered intratumorally into B6 mice bearing YUMM1.7 tumors (Fig. 2F). S1PR1 blockade significantly delayed tumor growth compared to vehicle-treated controls (Fig. 2G). Additionally, W146 treatment markedly reduced S1PR1 expression on intratumoral CD8⁺ T cells (Fig. EV3B). Further analysis revealed that the improved tumor control was associated with enhanced effector cytokine production (Fig. 2H) and decreased expression of exhaustion markers in intratumoral CD8⁺ T cells (Fig. 2I,J).

Together, these findings establish that disruption of the S1P-S1PR1 axis, either via SphK1 ablation or local pharmacological inhibition of S1PR1, revitalizes CD8⁺ T cell effector function and mitigates exhaustion, thereby enhancing anti-tumor immunity. These results underscore a pivotal role for S1P-S1PR1 signaling in promoting CD8⁺ T cell dysfunction in the TME.

## The S1P-S1PR1 signaling axis activates multiple stress response pathways in CD8⁺ T cells

To elucidate the mechanism by which S1PR1-mediated signaling compromises CD8⁺ T cell functionality and impairs anti-tumor responses, we performed RNA sequencing (RNA-seq) analysis on CD8⁺ T cells activated for 3 days in the presence of either exogenous S1P or vehicle control. Principal component analysis (PCA) revealed a distinct transcriptomic profile in S1P-treated CD8⁺ T cells compared to vehicle-treated controls (Fig. 3A). Gene set enrichment analysis (GSEA) of differentially expressed genes (DEGs; log2FC ≥0.3, Padj ≤0.05) demonstrated significant enrichment of cellular stress response pathways, including the stress-activated MAPK cascade, unfolded protein response (UPR), and endoplasmic reticulum (ER) stress (Fig. 3B). Notably, TGF-β receptor and SMAD signaling pathways were also enriched in S1P-treated CD8⁺ T cells, consistent with previous studies showing that S1P can cross-activate TGF-β receptor signaling and mimic TGF-β-induced cellular responses (Xin et al, 2004).

Further analysis of stress response-associated genes, as well as those critical for CD8⁺ T cell effector function and metabolic fitness, revealed that S1P treatment broadly dysregulated key transcripts. Genes involved in T cell activation and cell cycle progression (e.g., *Xcl1, Cxcr3, Cdk1, Cdk7, Cdk8, Cdk10*), T cell memory formation (e.g., *Id3, Eomes, Cd27, Tcf7, Il27ra*), and effector function (*Ifng,*

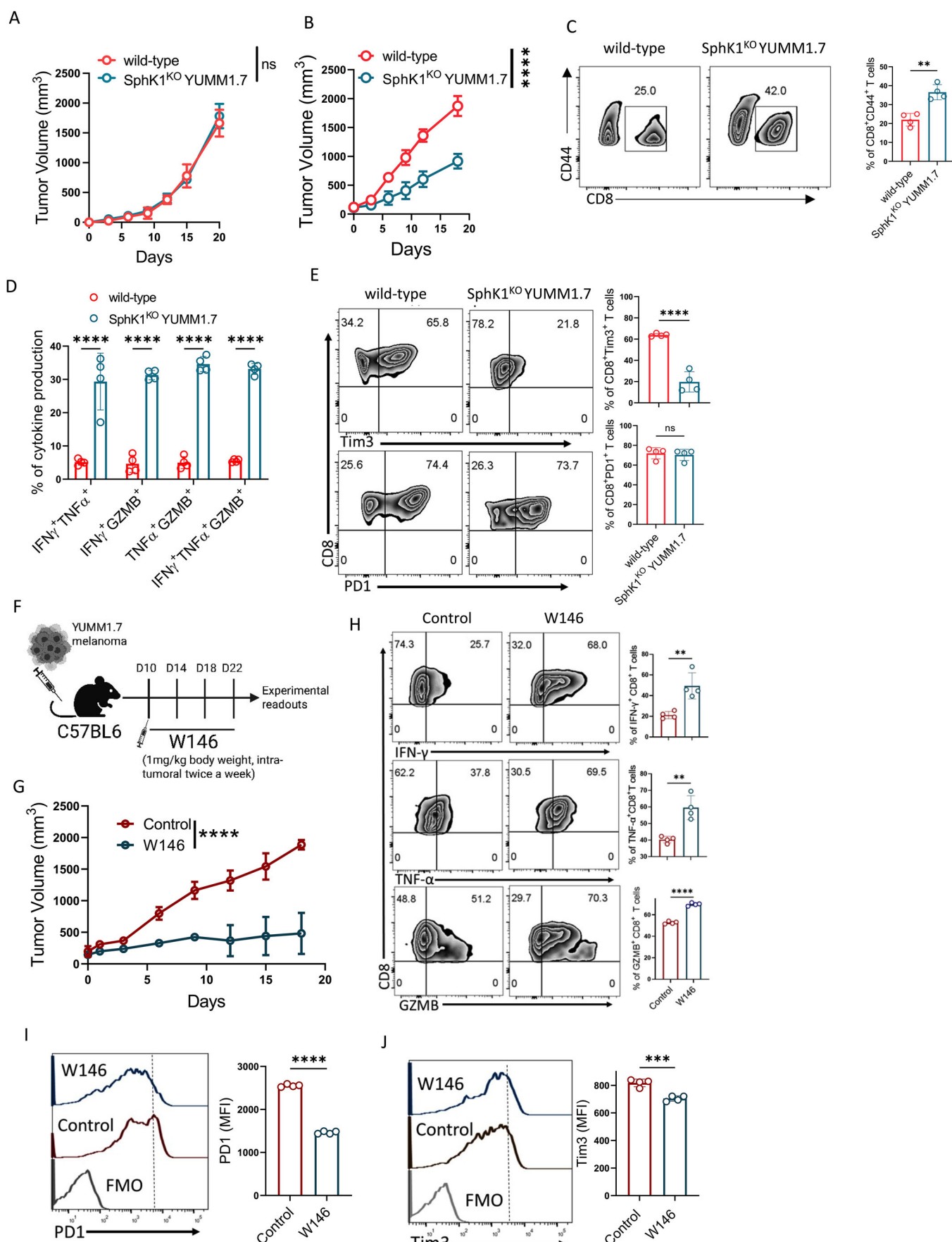

◀ **Figure 2. Disruption of the S1P-S1PR1 signaling axis enhances the anti-tumor function of CD8⁺ T cells.**

(A, B) Tumor progression was assessed in both (A) Rag1$^{-/-}$ ($n = 4$) and (B) immunocompetent C57BL/6 mice ($n = 4$) following subcutaneous implantation of either wild-type or SphK1$^{KO}$ YUMM1.7 melanoma cells. (C–E) CD8⁺ T cells obtained from the tumor sites of both wild-type and SphK1$^{KO}$ YUMM1.7 bearing mice were assessed for: (C) frequency of tumor antigen-experienced (CD8⁺CD44⁺) T cells ($n = 4$), (D) frequency of intratumoral CD8⁺ T cells co-producing two or more intracellular cytokines following re-stimulation with PMA + ionomycin ($n = 4$), and (E) expression of different exhaustion associated molecules ($n = 4$). (F–J) C57BL/6 mice ($n = 4$ mice/group) with subcutaneously established YUMM1.7 melanoma tumor treated either with vehicle control or W146, as (F) represented schematically, were evaluated for: (G) tumor growth, (H) the ability of CD8⁺ T cells from the tumor site to produce different effector cytokines upon re-stimulation in vitro, (I) expression of PD1 on intratumoral CD8⁺ T cells, and (J) expression of Tim3 on intratumoral CD8⁺ T cells. *$P < 0.05$; **$P < 0.01$; ***$P < 0.005$; ****$P < 0.0001$; ns, nonsignificant ($P > 0.05$), the error bar represents the standard deviation (SD). P values are derived from unpaired two-tailed Student's t test (C, E, H–J) and two-way Anova test (A, B, D, G). Source data are available online for this figure.

Il2, Tnf, Prf1) were significantly downregulated. In parallel, mitochondrial and TCA cycle-associated genes (e.g., Fasn, Atp5g1, Atp5g3, Acaca, Sdhb, Atp5e, Cox10, Cox19, Ndufs2, Ndufs3, Ndufs7), essential for maintaining metabolically fit and long-lived anti-tumor T cells, were also markedly suppressed (Fig. 3C).

Conversely, genes negatively associated with T cell function were enriched following S1P exposure. These included transcripts linked to MAP kinase signaling (e.g., Map3k15, Map3k13, Gadd45a, Gadd45b, Map3k8), apoptosis (e.g., Chac1, Trib3, Pmaip1, Cebpb, Fas, Casp4, Casp6), and exhaustion (Klf10, Tox, Ikzf2, Id2). Strikingly, S1P-treated CD8⁺ T cells also exhibited heightened expression of UPR-related genes (e.g., Atf3, Ero1a, Parp16, Eif2ak2, Optn, Creb3l2, Crebrf), including Ddit3 (encoding CHOP), a pivotal regulator of T cell dysfunction and impaired anti-tumor responses (Fig. 3C).

Together, these findings highlight a critical role for S1P-S1PR1 signaling in rewiring CD8⁺ T cell transcriptional programs to drive metabolic and functional dysregulation, ultimately compromising their anti-tumor potential.

## The S1P-S1PR1 signaling axis activates CHOP, resulting in CD8⁺ T cell death, impaired effector function, and diminished anti-tumor activity

Hyperactivation of UPR pathways has been implicated in driving T cell dysfunction in tumors (Cao et al, 2019; Song et al, 2018). Notably, CHOP, a downstream effector of the ER stress sensor PERK, was found to be upregulated in intratumoral CD8⁺ T cells, correlating with reduced effector function (Cao et al, 2019). Given the enriched expression of Ddit3 (encoding CHOP) in S1P-exposed CD8⁺ T cells, we examined whether S1P-S1PR1-driven dysfunction is mediated, at least in part, via CHOP activation. We first assessed CHOP levels in CD8⁺ T cells treated with exogenous S1P and observed a significant increase. This upregulation was largely PERK-dependent, as pharmacological inhibition of PERK using GSK2656157 (hereafter GSK) markedly reduced S1P-induced CHOP expression (Fig. 4A,B). To further confirm the role of the S1P-S1PR1 axis, we performed genetic knockdown of S1PR1 in CD8⁺ T cells using both siRNA and shRNA approaches. In both cases, S1P treatment failed to induce CHOP expression in S1PR1-knockdown CD8⁺ T cells (Fig. 4C,D).

Recognizing the critical role of the S1P-S1PR1 axis in CHOP activation, we next asked whether CHOP inhibition could rescue CD8⁺ T cell function. Indeed, pharmacological inhibition of CHOP activation using GSK restored effector cytokine production in S1P-treated CD8⁺ T cells (Fig. 4E). Strikingly, CHOP inhibition also improved CD8⁺ T cell viability following TCR re-stimulation, which

was otherwise markedly reduced upon S1P exposure (Fig. 4F). These findings suggest that S1P-S1PR1-driven CHOP activation may underlie the reduced effector function and heightened apoptotic susceptibility of tumor-reactive CD8⁺ T cells in the TME.

Finally, to determine whether CHOP inhibition can overcome S1P-S1PR1-mediated CD8⁺ T cell dysfunction in tumors, we subcutaneously implanted YUMM1.7 melanoma cells into B6 mice and treated them with either GSK or vehicle control (Fig. 4G). GSK treatment significantly delayed tumor growth compared to controls (Fig. 4H). Further analysis revealed a marked improvement in the effector function of intratumoral CD8⁺ T cells following GSK treatment, as evidenced by enhanced production of TNFα and GZMB upon TCR re-stimulation (Fig. 4I). Notably, the frequency of antigen-experienced CD8⁺ T cells was significantly higher in the GSK-treated group, suggesting that CHOP activation downstream of S1P-S1PR1 contributes to T cell attrition (Fig. 4J). Additionally, CHOP inhibition substantially reduced the expression of key exhaustion markers, mirroring the effects observed upon disruption of S1P-S1PR1 signaling in intratumoral CD8⁺ T cells (Fig. 4K,L).

Together, these findings position CHOP as a critical downstream effector of the S1P-S1PR1 signaling axis and a key regulator that not only impairs T cell effector function but also potentially promotes CD8⁺ T cell death in the TME.

## CHOP-mediated activation of p38MAPK promotes CD8⁺ T cell death in S1P-treated cells

Previous studies have identified CHOP as a transcription factor that regulates a broad array of genes involved in diverse cellular pathways (DeZwaan-McCabe et al, 2013; Goodall et al, 2010; Yang et al, 2017). Given that tumor antigen-reactive T cells often fail to mount an effective anti-tumor response due to rapid attrition at the tumor site, and considering that CHOP inhibition can partially restore CD8⁺ T cell viability, we next sought to identify the pathway mediating CHOP-induced T cell death. The p38MAPK pathway is known to drive T cell apoptosis and compromise their anti-tumor efficacy (Gurusamy et al, 2020). Consistent with this, our transcriptomic analysis revealed enrichment of MAP kinase pathway genes among the top 20 pathways in S1P-treated CD8⁺ T cells. We therefore assessed p38MAPK activation following S1P exposure. Indeed, S1P treatment during T cell activation led to a significant increase in phosphorylated p38MAPK levels, without altering total p38MAPK protein levels (Fig. 5A). In contrast, phosphorylation of other MAP kinases, such as p-ERK and p-JNK, remained unchanged between S1P-treated and vehicle control cells (Fig. EV4A,B). Notably, this increase in phospho-p38MAPK was abrogated by both siRNA and shRNA-mediated knockdown of

A

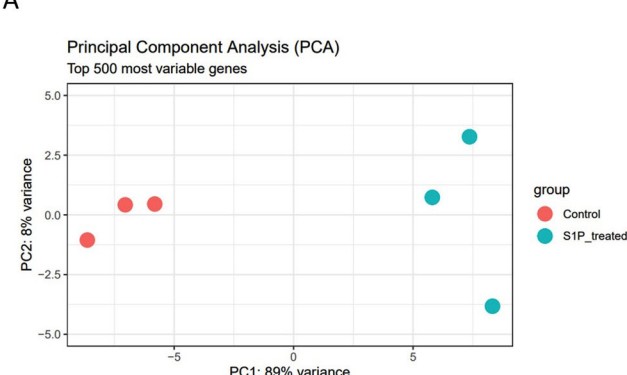

Principal Component Analysis (PCA)
Top 500 most variable genes

B

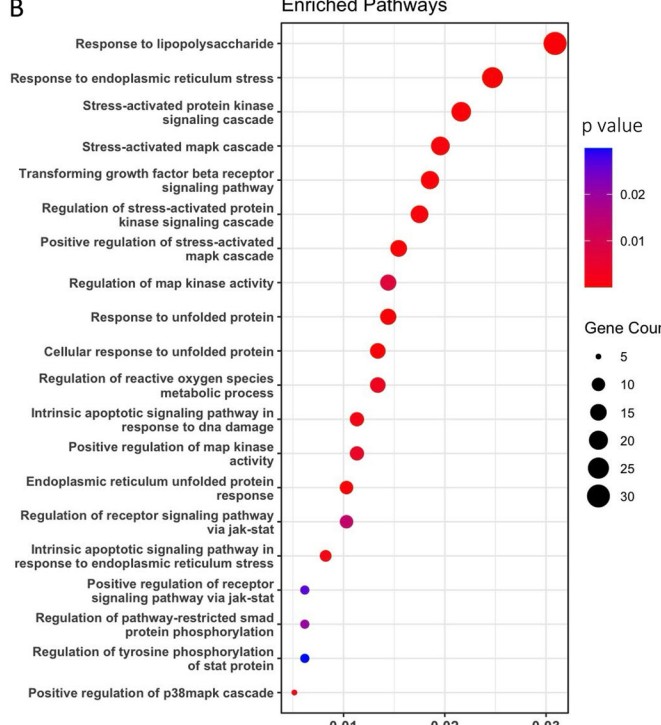

Enriched Pathways

C

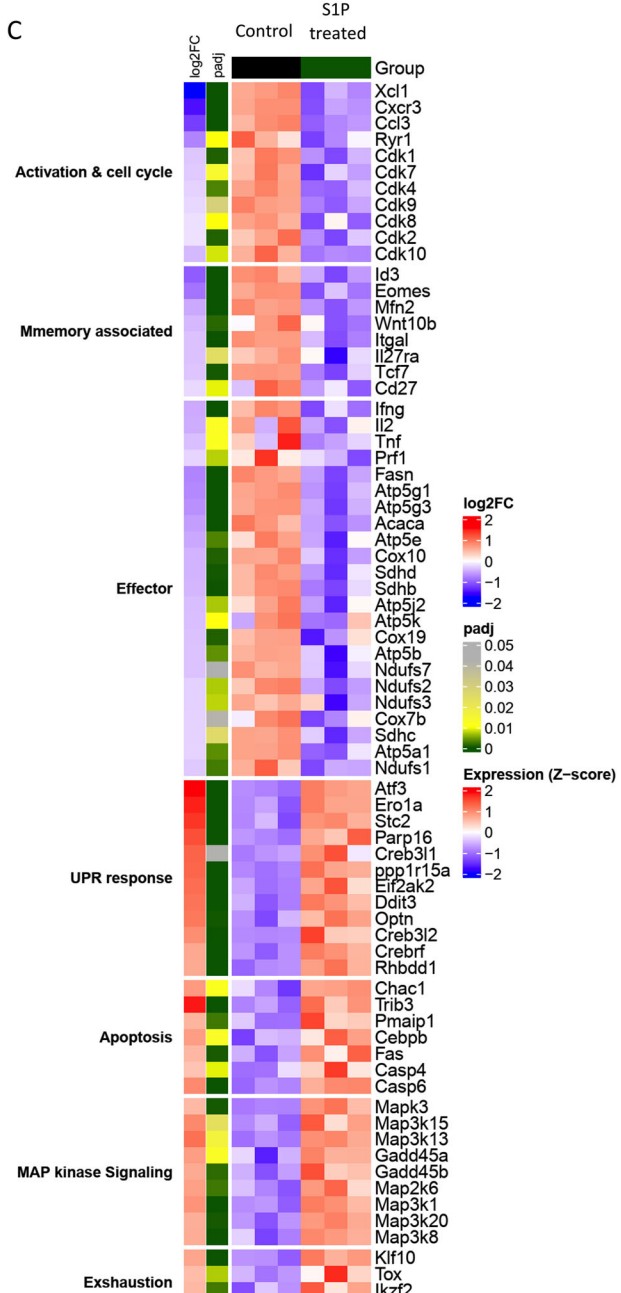

Figure 3. S1P-S1PR1 signaling rewires the transcriptional landscape of CD8$^+$ T cells by inducing different stress response pathways.

(A) Principal component analysis (PCA) plot depicting the distribution of the gene profile of each sample from two groups. (B) Gene set enrichment analysis (GSEA) showing the top 20 upregulated pathways based on differential gene expression (DEGs) upon S1P treatment. GO enrichment analysis was performed using enrichGO (clusterProfiler; over-representation analysis with a hypergeometric test; P values). (C) A comprehensive heatmap illustrates key DEGs (log2 fold change >0.3, Padj <0.05) involved in specific biological pathways, highlighting transcriptional differences between S1P-treated and vehicle-treated groups ($n = 3$ per group).

*S1pr1*, confirming its dependence on the S1P-S1PR1 axis (Fig. 5B,C).

To determine whether CHOP mediates this p38MAPK activation, we inhibited CHOP using GSK and observed a marked reduction in phospho-p38MAPK levels in S1P-treated CD8$^+$ T cells, with no effect on total p38MAPK expression (Fig. 5D). This suggests that CHOP may regulate phospho-p38MAPK via an upstream kinase necessary for its phosphorylation.

MAP kinase phosphorylation is typically driven by a hierarchical cascade, with MAP Kinase Kinase Kinase (MAP3K) phosphorylating MAP Kinase Kinase (MAP2K), which then activates MAP Kinase (MAPK). To identify the upstream kinase regulated by CHOP, we analyzed the transcript levels of various *Map3k* and *Map2k* genes that were found to be enriched in S1P-treated CD8$^+$ T cells based on our RNA-seq analysis. Compared to vehicle controls, S1P treatment significantly upregulated three *Map3k* transcripts- *Map3k8*, *Map3k13*, and *Map3k15*- whose expression was markedly reduced upon CHOP inhibition (Fig. 5E). In contrast, although *Map2k3*, *Map2k4*, and *Map2k6* were also upregulated by S1P, their expression remained unaffected by CHOP inhibition, suggesting that CHOP specifically regulates *Map3k* but not *Map2k* genes (Fig. 5F).

To further investigate whether CHOP directly regulates *Map3k* genes, we analyzed the promoters of *Map3k8*, *Map3k13*, and *Map3k15*, which were significantly upregulated upon S1P treatment. We identified potential CHOP-binding sites within these regions. Among these, *Map3k13* had the highest predicted CHOP-binding score, followed by *Map3k15* and *Map3k8* (Fig. EV4C). Consistent with these predictions, chromatin immunoprecipitation (ChIP) assays confirmed significantly increased CHOP binding to the *Map3k13* and *Map3k15* promoters, along with a modest increase in CHOP binding at the *Map3k8* promoter in S1P-treated CD8$^+$ T cells, comparable to the positive control gene *Dr5* (Fig. 5G). In contrast, no CHOP binding was observed at the promoter regions of *Map2k3*, *Map2k4*, *Map2k6*, or *Mapk14* (which encodes p38MAPK), reinforcing the conclusion that CHOP selectively enhances *Map3k* expression and promotes downstream phospho-p38MAPK signaling (Fig. 5G).

To assess the functional consequences of p38MAPK activation, we pre-treated CD8$^+$ T cells with the p38MAPK inhibitor SB203580 (p38i) prior to S1P exposure. Inhibition of p38MAPK significantly rescued CD8$^+$ T cell viability, confirming its role in S1P-induced cell death (Fig. 5H). Interestingly, p38MAPK inhibition also improved the cytokine production by CD8$^+$ T cells (Fig. 5I), as reported previously (Gurusamy et al, 2020). Moreover, p38MAPK inhibition enhanced mitochondrial metabolism in S1P-treated T cells, as reflected by increased oxygen consumption rate (OCR) (Fig. 5J).

Collectively, our findings show that CHOP, acting downstream of S1P-S1PR1 signaling, binds, at least in part, to the promoter regions of *Map3k13* and *Map3k15* to transcriptionally upregulate their expression. This, in turn, drives p38MAPK phosphorylation and promotes CD8$^+$ T cell death.

## p38MAPK inhibition revitalizes CD8$^+$ T cells and improves ACT response to anti-PD1 therapy

Having identified p38MAPK as a key downstream target of S1P-S1PR1-CHOP signaling that regulates CD8$^+$ T cell viability, we hypothesized that inhibiting p38MAPK would improve T cell persistence in tumors and augment anti-tumor immunity. To test this, we first examined whether intratumoral CD8$^+$ T cells exhibit elevated levels of phospho-p38MAPK. Indeed, CD8$^+$ T cells isolated from the tumors of B6 mice bearing subcutaneous YUMM1.7 melanoma showed significantly higher phospho-p38MAPK expression compared to splenic CD8$^+$ T cells (Fig. EV5A).

Next, we investigated whether p38MAPK inhibition enhances CD8$^+$ T cell persistence and anti-tumor responses. B6 mice bearing YUMM1.7 tumors were treated with a p38i or vehicle control (Fig. 6A). p38i treatment significantly delayed tumor growth and increased the frequency of intratumoral CD8$^+$ T cells compared to controls (Fig. 6B,C). Notably, in addition to improving T cell persistence, p38i also enhanced CD8$^+$ T cell effector function (Fig. 6D). This was accompanied by a marked reduction in the frequency of terminally exhausted CD8$^+$ T cells (PD1$^+$Tim3$^+$), suggesting that p38MAPK inhibition may mitigate T cell exhaustion within the TME (Fig. EV5B).

Given that resistance to anti-PD1 therapy often stems from the limited availability of tumor-specific T cells, we hypothesized that combining p38i with anti-PD1 could enhance therapeutic efficacy by preserving CD8$^+$ T cells within the TME. To test this, we employed the B16F10 melanoma model, which is refractory to anti-PD1 monotherapy. B6 mice were subcutaneously engrafted with B16F10 tumors and adoptively transferred with gp100-specific pmel-1 CD8$^+$ T cells prior to treatment initiation, allowing us to specifically assess the effects on tumor-reactive T cells (Fig. 6E). As expected, anti-PD1 monotherapy, with or without adoptive T cell transfer, failed to control tumor growth. In contrast, p38i treatment alone significantly delayed tumor progression in mice receiving pmel-1 T cells. Strikingly, the combination of p38i and anti-PD1 elicited a robust anti-tumor response in this otherwise resistant model (Fig. 6F). Analysis of intratumoral gp100-specific CD8$^+$ T cells (Vβ13$^+$CD8$^+$ T cells) revealed approximately a twofold increase in frequency following p38i treatment compared to vehicle controls. This increase was further amplified by anti-PD1 therapy, consistent with a PD1-blockade-mediated proliferation burst of tumor-specific T cells (Fig. 6G). Furthermore, the effector function of Vβ13$^+$CD8$^+$ T cells, as measured by production of IFN-γ, TNF-α, and granzyme B, was significantly enhanced by p38i treatment and further improved by the addition of anti-PD1 (Fig. 6H).

Given that p38MAPK activation is driven upstream by PERK-mediated activation of CHOP, we tested whether directly targeting PERK would similarly improve CD8$^+$ T cell function and responsiveness to anti-PD1 therapy. Adoptive transfer of Pmel-1

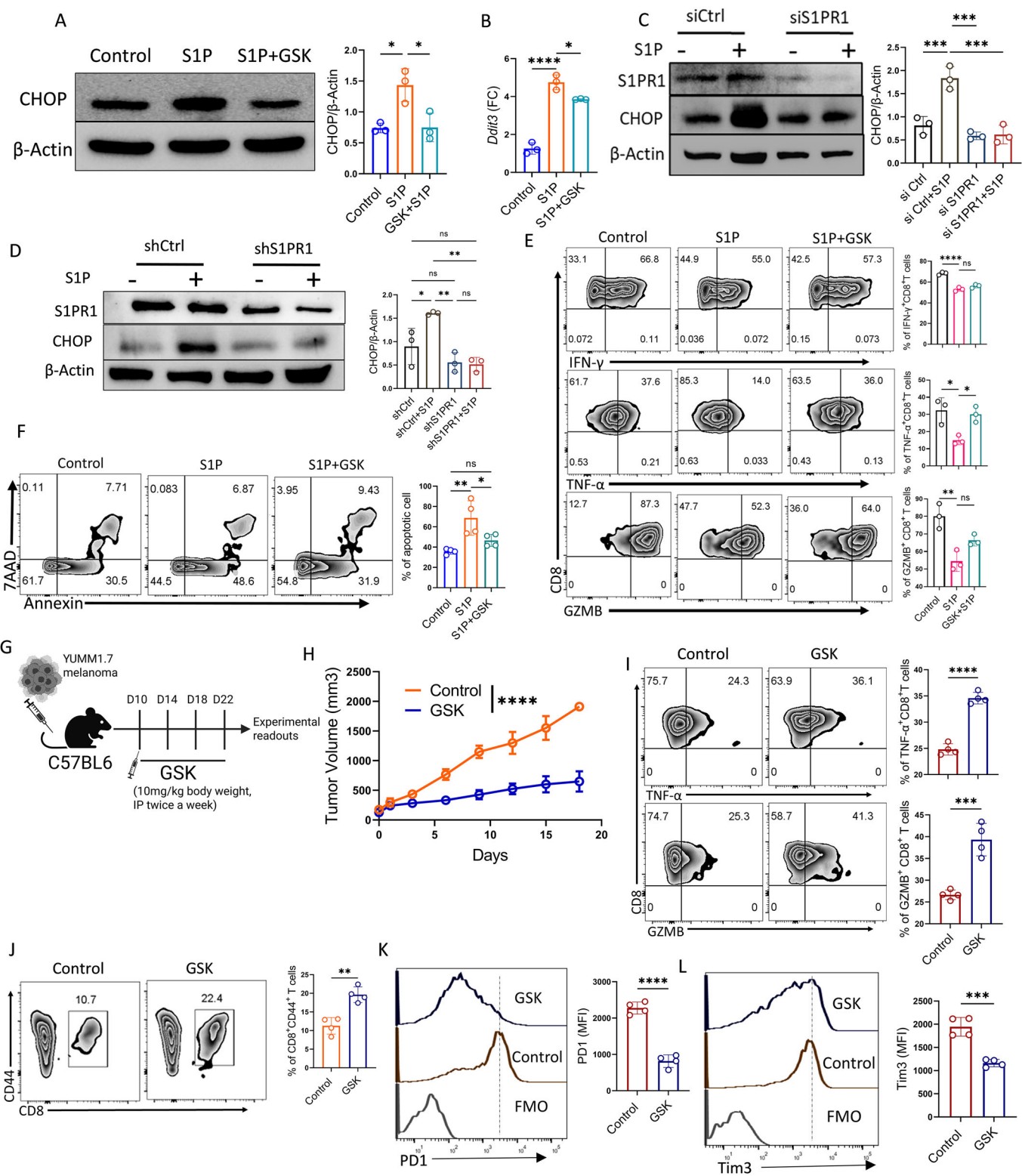

CD8+ T cells transduced with shRNA against PERK (referred to as Pmel^shPERK) significantly delayed B16F10 tumor growth relative to mice receiving Pmel-1 T cells transduced with control shRNA (referred to as Pmel^WT) (Fig. 6I,J). Remarkably, anti-PD1 therapy further boosted the anti-tumor activity of Pmel^shPERK cells, whereas

Pmel^WT cells remained unresponsive to checkpoint blockade. The enhanced responsiveness of Pmel^shPERK cells was associated with superior persistence in the TME, reduced exhaustion, and greater effector functionality of adoptively transferred Vβ13+CD8+ T cells compared to controls (Figs. 6K,L and EV5C).

**Figure 4. S1P-S1PR1 signaling induces CHOP expression, leading to CD8+ T cell dysfunction and death.**

(A) Western blot analysis showing expression of CHOP in control, S1P, and GSK pretreated S1P-treated CD8+ T cells. The adjacent bar graph depicts normalized densitometric data from three biological replicates ($n = 3$). (B) q-PCR analysis of *Ddit3* (encoding CHOP) in respective groups ($n = 3$). (C, D) Western blot analysis showing expression of CHOP in activated CD8+ T cells upon *S1pr1* knockdown using (C) siRNA and (D) shRNA. The adjacent bar graph depicts normalized densitometric data from three biological replicates ($N = 3$, for both (C, D). (E) Purified mouse CD8+ T cells activated under the indicated treatment conditions were analyzed for the production of effector cytokines. The adjacent bar plots represent cumulative data from three biological replicates ($n = 3$). (F) Purified mouse CD8+ T cells activated under the indicated treatment conditions were assessed for the frequency of CD8+ T cells undergoing apoptosis, as determined by Annexin V and 7AAD staining. The adjacent bar plots represent cumulative data from four biological replicates ($n = 4$). (G–L) C57BL/6 mice ($n = 4$ mice/group) with subcutaneously established YUMM1.7 melanoma tumor treated either with vehicle control or GSK, as (G) represented schematically, were evaluated for: (H) tumor growth, (I) the ability of CD8+ T cells from the tumor site to produce different effector cytokines, (J) frequency of CD8+ T cells at the tumor site, (K) expression of PD1, and (L) expression of Tim3 on intratumoral CD8+ T cells. *$P < 0.05$; **$P < 0.01$; ***$P < 0.005$; ****$P < 0.0001$; ns, nonsignificant ($P > 0.05$), the error bar represents the standard deviation (SD). $P$ values are derived from unpaired two-tailed Student's $t$ test (I–L), one-way ANOVA (A–F), and two-way ANOVA test (H). Source data are available online for this figure.

Together, these results reveal p38MAPK as a critical mediator of stress-driven T cell dysfunction in tumors. Inhibiting p38MAPK not only sustains CD8+ T cell survival and effector activity but also preserves a less exhausted T cell pool capable of responding to PD1 checkpoint blockade.

## PERK-CHOP-p38MAPK inhibition overcomes resistance to anti-PD1 therapy

To extend our adoptive transfer findings into a broader therapeutic context, we next evaluated whether pharmacologic inhibition of either PERK-mediated activation of CHOP or p38MAPK could enhance responses to checkpoint therapy in immunotherapy-refractory B16F10 melanoma tumors. Mice subcutaneously established with B16F10 melanoma tumors were treated with either a p38MAPK inhibitor (p38i) or a PERK-mediated CHOP activation inhibitor (GSK), alone or in combination with anti-PD1 (Fig. 7A). Both p38i and GSK monotherapies significantly delayed tumor progression, and importantly, their combination with anti-PD1 further enhanced tumor control beyond either agent alone (Fig. 7B).

Analysis of intratumoral T cells demonstrated an increased accumulation of CD8+ T cells in mice treated with p38i and GSK, suggesting improved persistence of effector populations (Fig. 7C). This effect was further amplified when combined with anti-PD1, consistent with an expanded pool of immunologically active CD8+ T cells. Notably, p38i treatment markedly reduced the proportion of terminally exhausted PD1+Tim3+ CD8+ T cells, whereas GSK treatment resulted in a modest decrease in this population (Fig. 7D). Moreover, CD8+ T cells from treated tumors displayed enhanced effector cytokine production (IFN-γ, TNF-α, and GZMB), demonstrating improved functional competence (Fig. 7E).

These data identify the PERK-CHOP-p38MAPK pathway as a clinically actionable target to overcome resistance to immunotherapy.

## Discussion

The functionality of tumor-specific CD8+ T cells within the tumor microenvironment (TME) is constrained by various microenvironmental cues that collectively induce T cell dysfunction, ultimately compromising the anti-tumor immune response (Anderson et al, 2017). Over the past few decades, the discovery of regulatory molecules, such as PD-1, CTLA-4, and Tim-3, that suppress T cell effector functions in the TME has significantly deepened our understanding of T cell exhaustion in cancer and laid the foundation for the development of immune checkpoint blockade (ICB) therapies (Pardoll, 2012). However, recent studies highlighting the circumscribed efficacy of ICB in only a subset of patients indicate the involvement of additional regulatory mechanisms that contribute to distinct aspects of T cell exhaustion beyond classical immune checkpoints. In this context, the present study uncovers a previously underexplored yet critical role of the S1P-S1PR1 axis in influencing the attrition of tumor-specific CD8+ T cells in the tumor by increasing their susceptibility to cell death, a critical hallmark of exhausted T cells. We demonstrate that S1PR1, highly expressed on intratumoral tumor antigen-specific CD8+ T cells, engages with its ligand S1P to induce phosphorylation of p38 MAPK through activation of the ER stress sensor CHOP, thereby exacerbating CD8+ T cell exhaustion and promoting their loss in the TME. Although S1PR1 is known to activate several canonical pathways, including RAS-MAPK, PI3K-AKT, STAT3, and ERK, our findings reveal that its pro-apoptotic activity in CD8+ T cells is predominantly mediated through PERK-dependent CHOP activation. This previously unrecognized link between S1PR1-induced ER stress and CD8+ T cell attrition provides novel mechanistic insight into tumor-induced T cell exhaustion. Consistent with this model, pharmacological inhibition of either PERK-mediated CHOP activation or p38 MAPK substantially enhanced the persistence of CD8+ T cells in tumors and markedly improved their responsiveness to anti-PD-1 therapy.

Beyond driving effector CD8+ T cell dysfunction, S1PR1 also plays a pivotal role in inversely regulating the development of tissue-resident memory T ($T_{RM}$) cells (Mackay et al, 2013; Skon et al, 2013; Zhu et al, 2013). Because $T_{RM}$ cells characteristically express low levels of S1PR1, our findings suggest that they may be inherently protected from S1P-mediated suppression in the S1P-rich tumor microenvironment. This reduced susceptibility could contribute to the favorable association between $T_{RM}$ abundance and improved tumor control (Gavil et al, 2024). However, it remains unclear whether $T_{RM}$ cells, upon re-encounter with antigen, generate effector progeny that re-express KLF2 and S1PR1, and thus become vulnerable to S1P-driven dysfunction, similarly observed in peripherally recruited tumor-reactive CD8+ T cells. These insights raise the possibility that therapeutic inhibition of S1PR1 or its downstream signaling could simultaneously reinvigorate exhausted effector CD8+ T cells while preserving the functional integrity of $T_{RM}$ cells, thereby enhancing anti-tumor immunity on multiple fronts.

The expression of S1PR1 on various immune cell types in TME, such as Tregs and MDSCs, as well as on tumor cells, is well documented (Priceman et al, 2014; Zhou and Guo, 2018). In this

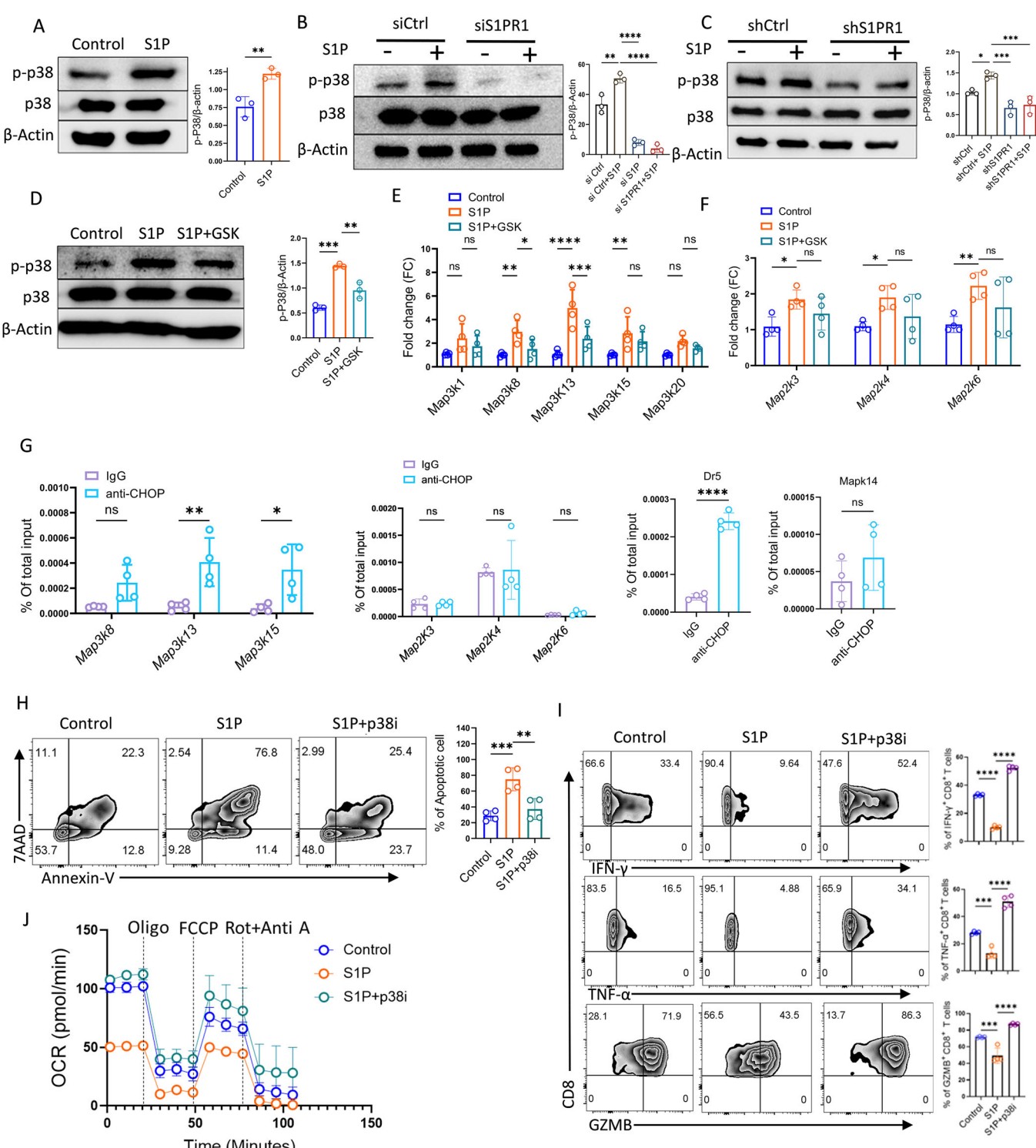

study, we show that intratumoral CD8+ T cells, particularly those characterized as a terminally exhausted subset that fail to respond to ICB therapy, exhibit elevated levels of S1PR1. We propose that this upregulation is likely driven by sustained activity of STAT3, a transcription factor known to directly regulate the *S1pr1* gene and commonly associated with dysfunctional CD8+ T cells (Lee et al, 2010; Zhang et al, 2020). In fact, in our study, we also observed that

the intratumoral CD8+ T cells expressing S1PR1 had elevated levels of phospho-STAT3 compared to S1PR1Lo CD8+ T cells.

Importantly, our findings reveal that S1PR1-mediated T cell dysfunction is predominantly initiated by activation of the S1P-S1PR1 signaling axis in CD8+ T cells in the tumor milieu. This notion was supported by our observation that genetic deletion of *Sphk1* in tumor cells abrogated the features of T cell exhaustion,

**Figure 5.  CHOP activates p38 MAPK to drive CD8⁺ T cell death.**

(A) Western blot analysis of phospho-p38 (p-p38) and total p38 expression, in vehicle control and S1P-treated CD8⁺ T cells. The adjacent bar graph depicts normalized densitometric data from three biological replicates ($n = 3$). (B, C) Western blot analysis showing the expression of p-p38 and total p38 in activated T cells upon *S1pr1* knockdown using (B) siRNA ($n = 3$) and (C) shRNA ($n = 3$). The adjacent bar graph depicts normalized densitometric data. (D) Western blot analysis of p-p38 and total p38 in CD8⁺ T cells activated in the presence or absence of S1P, along with the indicated inhibitor. The adjacent bar graph depicts normalized densitometric data from three biological replicates ($n = 3$). (E, F) qPCR analysis of transcript levels of different (E) *Map3k* and (F) *Map2k* genes in CD8⁺ T cells in respective groups ($n = 4$). (G) CD8⁺ T cells were activated in the presence or absence of S1P and were collected and processed for chromatin-immunoprecipitation (ChIP) assay with an antibody specific for CHOP or with rabbit IgG control. qPCR primers specific for the known CHOP binding gene (*Dr5*) and different *Map3K* and *Map2K*, along with *Mapk14*, were used to determine CHOP binding to the respective promoters ($n = 4$). (H, I) Purified mouse CD8⁺ T cells activated under the indicated treatment conditions were assessed for: (H) T cell death by Annexin V and 7AAD staining and (I) frequency of CD8⁺ T cells producing different effector cytokines. The adjacent bar represents cumulative data from four biological replicates ($n = 4$, for both (H, I)). (J) Extracellular flux assay for determining of oxygen consumption rate (OCR) in activated CD8⁺ T cells in respective groups. *$P < 0.05$; **$P < 0.01$; ***$P < 0.005$; ****$P < 0.0001$; ns, nonsignificant ($P > 0.05$), the error bar represents the standard deviation (SD). *P* values are derived from unpaired two-tailed Student's *t* test (A), one-way ANOVA (B–D, H, I), and two-way ANOVA test (E–G). Source data are available online for this figure.

presumably due to disruption of S1P-S1PR1 signaling. Congruent with previous reports, we found that tumor cells are the primary source of S1P, owing to their elevated expression of SphK1 (Wang et al, 2020). The critical role of extracellular S1P in promoting CD8⁺ T cell dysfunction through this axis was corroborated by in vitro studies in which exogenous S1P was added during CD8⁺ T cell activation. We observed that while naive CD8⁺ T cells express minimal S1PR1, likely essential for their retention in secondary lymphoid organs, S1PR1 expression rapidly increased following TCR stimulation. Notably, the addition of exogenous S1P to CD8⁺ T cells after 24 h of TCR stimulation led to metabolic and functional aberration without impacting cell viability. It is also noteworthy that although no overt loss of viability was observed in activated CD8⁺ T cells treated with exogenous S1P, the S1P-S1PR1 axis rendered them more susceptible to apoptosis following subsequent TCR stimulation. This pro-apoptotic effect stands in sharp contrast to findings in naive T cells, where S1P-S1PR1 signaling has been shown to promote cell survival by limiting JNK activation and maintaining a favorable balance of BCL-2 family proteins (Dixit et al, 2024). We propose that this discrepancy reflects fundamental differences in the intracellular signaling networks (signalomes) that regulate naive versus activated T cells, resulting in distinct downstream consequences of S1P-S1PR1 engagement. Additionally, the differential expression levels of S1PR1 between naive and activated T cells may further shape the signaling cascades induced by S1P, ultimately determining the opposing survival outcomes observed.

Elevated CHOP levels have been consistently detected in intratumoral CD8⁺ T cells across various preclinical and clinical tumor samples, and are associated with poor clinical outcomes in advanced ovarian cancer (Cao et al, 2019). CHOP, a downstream effector of the UPR sensor PERK, is a known mediator of anti-tumor CD8⁺ T cell impairment (Cao et al, 2019). However, the upstream signals responsible for its activation in the TME have been unclear. Our findings suggest that the S1P-S1PR1 signaling axis contributes, at least in part, to the sustained activation of CHOP in dysfunctional intratumoral CD8⁺ T cells. Importantly, the S1P-S1PR1 signaling pathway appears to curtail the anti-tumor potential of CD8⁺ T cells primarily through CHOP activation. That said, we cannot exclude the involvement of other pathways, particularly TGFβ receptor signaling, which was enriched in our RNA-seq analysis of S1P-treated CD8⁺ T cells and is previously implicated in T cell dysfunction (Chen, 2023; Thomas and Massague, 2005). Consistent with prior reports, our data demonstrate that activation of CHOP downstream of S1P-S1PR1

signaling diminishes the effector function of CD8⁺ T cells, as its inhibition restores cytokine production by CD8⁺ T cells (Cao et al, 2019). Intriguingly, beyond impairing effector functions, CHOP activation also contributes to the progressive loss of tumor-reactive CD8⁺ T cells in the TME, likely through engagement of pro-apoptotic pathways, as previously described (Oyadomari and Mori, 2004).

To elucidate downstream effectors of CHOP, we investigated p38MAPK, which emerged as a key candidate from our transcriptomic analyses. In agreement with recent findings, inhibition of p38MAPK markedly enhanced the anti-tumor capacity of CD8⁺ T cells (Gurusamy et al, 2020). Most notably, our results reveal a previously underappreciated role of CHOP in driving p38MAPK activation, which promotes apoptosis in S1PR1-expressing CD8⁺ T cells and limits their persistence in the TME. Delving deeper into this axis revealed that CHOP transcriptionally regulates p38MAPK signaling by binding to the promoter region of *Map3k13* and *Map3k15* transcripts. Our study offers compelling experimental evidence that p38MAPK activation is a key barrier to the persistence of tumor-reactive CD8⁺ T cells in the TME. Targeting this pathway not only enriches CD8⁺ T cell abundance but also enhances responsiveness to anti-PD-1 therapy. We propose that the mechanism underlying this benefit is linked to the known role of p38MAPK in promoting T cell death, potentially through the compromise of cellular antioxidant defences, as shown previously (Gurusamy et al, 2020). However, the exact mechanism requires further investigation.

In summary, our study identifies the S1P-S1PR1-CHOP-p38 MAPK axis as a previously unrecognized regulatory circuit that contributes to CD8⁺ T cell dysfunction and attrition in tumors. These findings uncover new mechanistic underpinnings of T cell exhaustion and reveal p38 MAPK as a promising therapeutic target to enhance CD8⁺ T cell survival and responsiveness to ICB therapies in the TME. While S1PR1 has been primarily associated with activation of signaling pathways such as STAT3, PI3K-AKT, and ERK that promote T-cell survival, proliferation, and trafficking, our data uncover a distinct, stress-associated S1PR1-PERK-CHOP-p38 MAPK axis that emerges under the chronic S1P exposure characteristic of the TME. Selectively targeting this maladaptive signaling state may enhance therapeutic specificity by sparing beneficial S1PR1 functions and reducing systemic toxicity associated with global receptor inhibition. Collectively, these insights highlight the novelty and translational potential of modulating S1PR1-driven stress signaling to reinvigorate anti-tumor CD8⁺ T cell immunity.

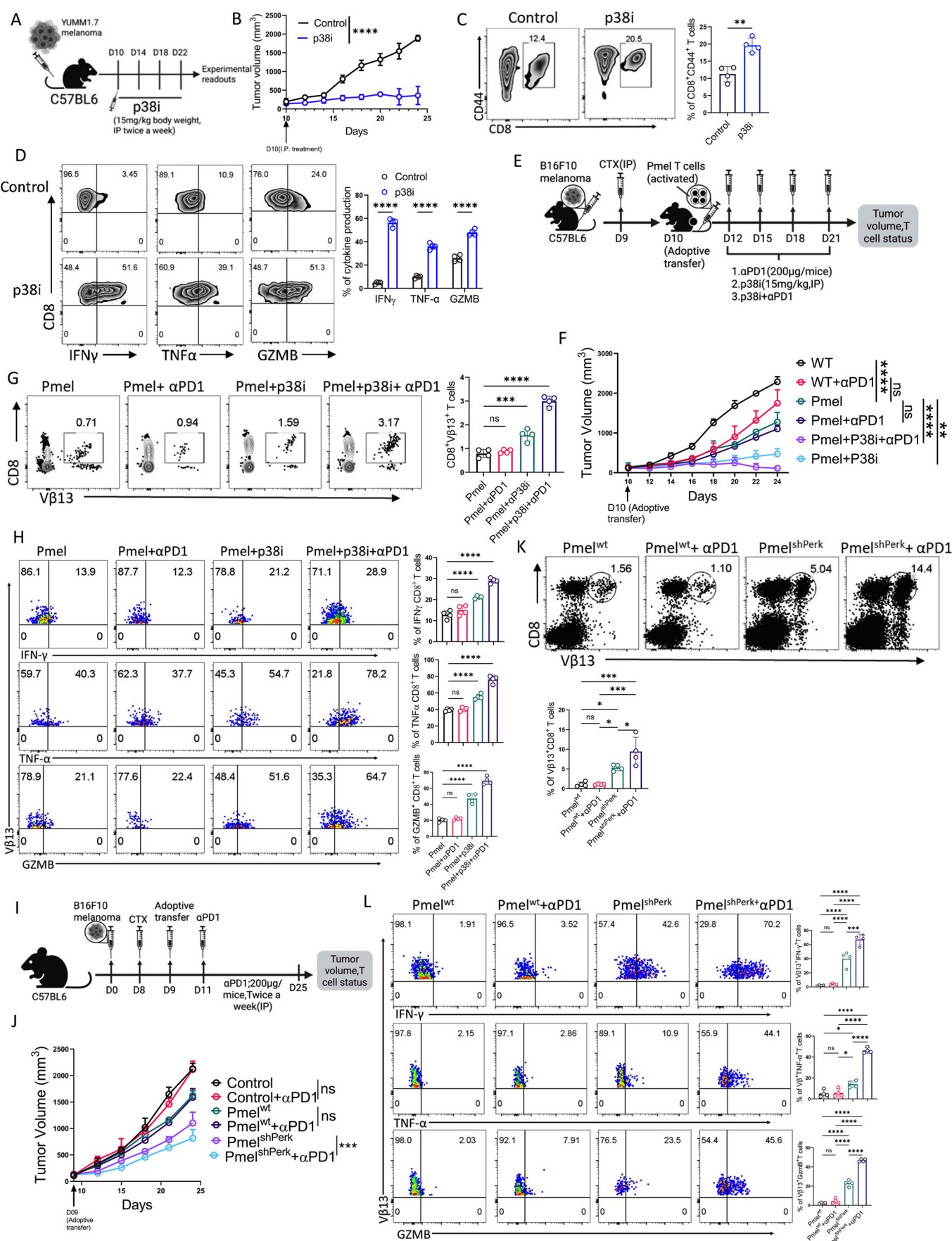

◀ **Figure 6. p38MAPK inhibition restores CD8⁺ T cell function and boosts ACT response to anti-PD-1 efficacy.**

(A–D) C57BL/6 mice ($n = 4$ mice/group) with subcutaneously established YUMM1.7 melanoma tumor treated either with vehicle control or p38i, as (A) represented schematically, were evaluated for: (B) tumor growth, (C) frequency of CD8⁺ T cells at the tumor site, (D) the ability of CD8⁺ T cells from the tumor site to produce different effector cytokines. (E) Schematic representation of the ACT protocol where C57BL/6 mice ($n = 4$ mice/group) bearing subcutaneous B16-F10 tumors were adoptively transferred with $0.75 \times 10^6$ Pmel-1 T cells, followed by treatment with or without anti-PD1 antibody (Clone# RMP1-14; 200 µg/mouse twice weekly), combined with p38i or vehicle control. Mice were subsequently evaluated for: (F) tumor growth ($n = 4$), (G) frequency of Vβ13⁺CD8⁺ T cells at the tumor site ($n = 4$), and (H) Intracellular expression of effector cytokines in intratumoral Pmel-1 T cells following in vitro restimulation ($n = 4$). (I) Schematic representation of the ACT protocol where C57BL/6 mice ($n = 4$ mice/group) bearing subcutaneous B16-F10 tumors were adoptively transferred with $0.75 \times 10^6$ Pmel-1 T cells transduced with either control shRNA (PmelWT) or shRNA targeting PERK (PmelPERK), followed by treatment with or without anti-PD1 antibody (Clone# RMP1-14; 200 µg/mouse twice weekly). Mice were evaluated for: (J) tumor growth ($n = 4$), (K) frequency of Vβ13⁺CD8⁺ T cells at the tumor site ($n = 4$), and (L) Intracellular expression of effector cytokines in intratumoral Pmel-1 T cells following in vitro restimulation ($n = 4$). *$P < 0.05$; **$P < 0.01$; ***$P < 0.005$; ****$P < 0.0001$; ns, nonsignificant ($P > 0.05$), the error bar represents the standard deviation (SD). $P$ values are derived from unpaired two-tailed Student's $t$ test (C, D), one-way ANOVA (G, H, K, L), and two-way ANOVA test (B, F, J). Source data are available online for this figure.

# Methods

### Reagents and tool table

| Reagent/resource | Reference or source | Identifier or catalog number |
|---|---|---|
| **Experimental models** | | |
| HEK-293Tcells | ATCC | CRL-3216 |
| Yumm1.7 cells | ATCC | CRL-3362 |
| B16-F10 Melanoma cells | ATCC | CRL-6475 |
| EL-4 Thymoma | ATCC | TIB-39 |
| Mouse: C57BL/6 J wildtype mouse | Jackson Laboratory | NA |
| Mouse: pmel-1 TCR transgenic mouse | Jackson Laboratory | NA |
| Mouse: Rag1 deficient mouse | Jackson Laboratory | NA |
| **Recombinant DNA** | | |
| psPAX2 | Addgene | 12260 |
| pMD2.G | Addgene | 12259 |
| Eif2ak3 Mouse shRNA Plasmid (Locus ID 13666, Vector: pGFP-C-shLenti) | Origene | TL510962 |
| S1pr1 Mouse shRNA Plasmid (Locus ID 13609, Vector: pGFP-C-shLenti) | Origene | TL516557 |
| S1pr1 siRNA | Thermo Scientific | AM16708 |
| **Antibodies** | | |
| Anti-mouse CD3 | BioXcell | Clone: 145-2C11; BE001-1 |
| Anti-mouse CD28 | BioXcell | Clone: 37.51; BE0015-1 |
| Anti-mouse CD4-PECY7 | Thermo Fisher Scientific | Clone: GK 1.5; 25-0041-82 |
| Anti-mouse CD8-AF-700 | Biolegend | Clone: 53-6.7; 100730 |
| Anti-mouse CD3-BV711 | BD Biosciences | Clone: 17A2; 740739 |

| Reagent/resource | Reference or source | Identifier or catalog number |
|---|---|---|
| Anti-mouse IFN-γ-APC | Thermo Fisher Scientific | Clone: XMG1.2; 17-7311-82 |
| Anti-mouse Granzyme-B-PE | Invitrogen | Clone: GB12; MHGB04 |
| Anti-mouse TNF-α-APC/CY7 | BD Biosciences | Clone: MP6-XT22; 560658 |
| Anti-mouse CD25-APC | BD Biosciences | Clone: PC61; 557658 |
| Anti-mouse/human CD44-PERCP/CY5.5 | Thermo Fisher Scientific | Clone: IM-7; 45-0441-82 |
| Anti-mouse Vβ13 TCR-FITC | BD Biosciences | MR12-3; 553204 |
| Anti-mouse Vβ13 TCR-PE | Biolegend | Clone: H132; 333108 |
| Anti-mouse CD8-APCCY7 | Molecular Probes | Clone: 53-6.7; A15316 |
| Anti-mouse CD4-BUV395 | BD Biosciences | Clone: GK1.5; 565974 |
| Anti-mouse S1PR1-PE | R&D | MAB7089 |
| Anti-mouse PD1-PECY7 | Biolegend | Clone: 9.3; BE0248 |
| Anti-mouse Tim3-BV421 | Biolegend | Clone: 5D12/TIM3; 747626 |
| Anti-human CD3 | BioXcell | Clone: OKT-3); BE001-2 |
| Anti-human CD28 | BioXcell | Clone: 9.3; BE0248 |
| Anti-human CD8-BV-786 | Biolegend | Clone: SK1; 344740 |
| Anti-human CD8-AF700 | BD Biosciences | Clone: OKT-8; 557945 |
| Anti-human PD1 PE-Cy7 | Biolegend | Clone: EH12.1; 561272 |
| Anti-human Tim3-BV421 | Biolegend | Clone: 7D3; 565562 |
| anti-human CD39 PE/Dazzle™ 594 | Biolegend | Clone: A1; 328224 |
| Anti-TOX Alexa Fluor™ 647 | BD Biosciences | Clone: NAN448B; 568356 |

| Reagent/resource | Reference or source | Identifier or catalog number |
|---|---|---|
| Anti-TOX -PE | Invitrogen | Clone: TXRX10; 12-6502-82 |
| Anti-human IFNγ-APC | Invitrogen | Clone: 4S.B3; 17-7319-82 |
| Anti-human TNF-α-percpcy5.5 | Biolegend | Clone: MAb11; 502926 |
| Anti rabbit IgG(H + L)- FITC | Invitrogen | A21206 |
| CHOP Mouse mAb | Cell Signaling Technologies | 2895 |
| p-P38 MAPK Rabbit mAb | Cell Signaling Technologies | 9211 |
| Total P38 MAPK Rabbit mAb | Cell Signaling Technologies | 9212 |
| β-actin Rabbit mAb | Cell Signaling Technologies | 4967 |
| S1PR1-Rabbit mAb | Cell Signaling Technologies | 63335 |
| SPHK1-Rabbit mAb | Cell Signaling Technologies | 12071 |
| **Oligonucleotides and sequence-based reagents** | | |
| Gene name | Primer | |
| *Map2k3* F | GCCTCAGACCAAAGGAAAATCC | IDT |
| *Map2k3* R | GGTGTGGGGTTGGACACAG | IDT |
| *Map2k4* F | AATCGACAGCACGGTTTACTC | IDT |
| *Map2k4* R | TGAAATCCCAGTGTTGTTCAGG | IDT |
| *Map2k6* F | ATGTCTCAGTCGAAAGGCAAG | IDT |
| *Map2k6* R | TTGGAGTCTAAATCCCGAGGC | IDT |
| *Mapk14* F | TGACCCTTATGACCAGTCCTTT | IDT |
| *Mapk14* R | GTCAGGCTCTTCCACTCATCTAT | IDT |
| *Map3k1* F | TAAATACCGGGTGTTTATTGGGC | IDT |
| *Map3k1* R | TTTTCTCCATAACATGGGGTCAG | IDT |
| *Map3k8* F | ATGGAGTACATGAGCACTGGA | IDT |
| *Map3k8* R | GGCTCTTCACTTGCATAAAGGTT | IDT |
| *Map3k13* F | CCCGACCTCATCTCCACAG | IDT |
| *Map3k13* R | TGGAAACAGGGATCATAGGGTT | IDT |
| *Map3k15* F | TGATGAACTGGCGAAAGAGTTG | IDT |
| *Map3k15* R | ATGATGATGTCCGATGTCAGAAC | IDT |
| *Map3k20* F | GAGATTGGTGCGTGGACTGAA | IDT |
| *Map3k20* R | CGCTTGCCTGTGATGTTGTT | IDT |
| *S1pr1* F | ATGGTGTCCACTAGCATCCC | IDT |
| *S1pr1* R | CGATGTTCAACTTGCCTGTGTAG | IDT |
| *S1pr2* F | ATGGGCGGCTTATACTCAGAG | IDT |
| *S1pr2* R | GCGCAGCACAAGATGATGAT | IDT |
| *S1pr3* F | ACTCTCCGGGAACATTACGAT | IDT |
| *S1pr3* R | CAAGACGATGAAGCTACAGGTG | IDT |
| *S1pr4* F | GTCAGGGACTCGTACCTTCCA | IDT |
| *S1pr4* R | GATGCAGCCATACACACGG | IDT |
| *S1pr5* F | GCTTTGGTTTGCGCGTGAG | IDT |
| *S1pr5* R | GGCGTCCTAAGCAGTTCCAG | IDT |
| *Glut1* F | CAGTTCGGCTATAACACTGGTG | IDT |

| Reagent/resource | Reference or source | Identifier or catalog number |
|---|---|---|
| *Glut1* R | GCCCCCGACAGAGAAGATG | IDT |
| *Hk2* F | GGAACCGCCTAGAAATCTCC | IDT |
| *Hk2* R | GGAGCTCAACCAAAACCAAG | IDT |
| *Pfk* F | AGGAGGGCAAAGGAGTGTTT | IDT |
| *Pfk* R | TTGGCAGAAATCTTGGTTCC | IDT |
| *Ldha* F | TGTCTCCAGCAAAGACTACTGT | IDT |
| *Ldha* R | GACTGTACTTGACAATGTTGGGA | IDT |
| *Enolase* F | AAAGATCTCTCTGGCGTGGA | IDT |
| *Enolase* R | CTTAACGCTCTCCTCGGTGT | IDT |
| *Opa* F | TGGAAAATGGTTCGAGAGTCAG | IDT |
| *Opa* R | CATTCCGTCTCTAGGTTAAAGCG | IDT |
| *Mfn-1* F | ATGGCAGAAACGGTATCTCCA | IDT |
| *Mfn-1* R | CTCGGATGCTATTCGATCAAGTT | IDT |
| *Mfn-2* F | CCAACTCCAAGTGTCCGCTC | IDT |
| *Mfn-2* R | GTCCAGCTCCGTGGTAACATC | IDT |
| *Drp1* F | CAGGAATTGTTACGGTTCCCTAA | IDT |
| *Drp1* R | CCTGAATTAACTTGTCCCGTGA | IDT |
| *β-Actin* F | ACGTAGCCATCCAGGCTGGTG | IDT |
| *β-Actin* R | TGGCGTGAGGGAGAGCAT | IDT |
| **Chemicals, enzymes, and other reagents** | | |
| RPMI-1640 medium (Glutamax) | Gibco | 61870-036 |
| FBS (Heat Inactivated – US origin) | Gibco | 10082147 |
| Penicillin-streptomycin solution | Himedia | A001 |
| Recombinant human IL-2 | R&D Scientific | BT-002-100 |
| Collagenase type IV | Himedia | TCL117 |
| 70 µm strainer | Falcon | 352350 |
| PMA | Cayman Chemical | 10008014 |
| Ionomycin | Cayman Chemical | 11312 |
| Golgi Plug | BD Biosciences | 555029 |
| BD Cytofix/Cytoperm Kit | BD Biosciences | 554714 |
| FoxP3 Staining Buffer Set | Invitrogen | 00-5523-00 |
| Seahorse XF Base Medium | Agilent Seahorse | 103335-100 |
| Seahrose XF Calibrant | Agilent Seahorse | 100840-000 |
| Sodium pyruvate | Gibco | 11360-070 |
| D-glucose | Cayman Chemicals | 11360-070 |
| Cell-Tak | Corning | 354240 |
| Oligomycin | Cayman Chemicals | 11341 |
| FCCP | Cayman Chemicals | 15218 |

| Reagent/resource | Reference or source | Identifier or catalog number |
|---|---|---|
| Rotenone | Cayman Chemicals | 13995 |
| Antimycin A | Cayman Chemicals | 19433 |
| 2-deoxy-D-glucose | Cayman Chemicals | 14325 |
| RIPA buffer | Himedia | TLC 121 |
| Protease and phosphatase inhibitor cocktails (PIC) | Himedia | ML051 |
| Bradford reagent | Himedia | ML106 |
| Laemmli buffer | Himedia | ML121 |
| PVDF membranes | Bio Rad | 1620112 |
| Clarity Enhanced chemiluminescence (ECL) | Bio-Rad | 1705062 |
| Dulbecco's modified Eagle's medium (DMEM; | Thermo Fisher Scientific | 10566-016 |
| Calcium Chloride | Himedia | MB034 |
| Polyethersulfone filter | Sartorius | 15406 |
| Lenti-X Concentrator | Takara | Cat no 631232 |
| RetroNectin | Takara | T100A |
| TRIzol reagent | Invitrogen | 15596026 |
| Nuclease free water | Himedia | TLC016 |
| iScript cDNA Synthesis Kit | Biorad | 1708890 |
| iTaq Universal SYBR Green Supermix | Biorad | 1725120 |
| BD Fixation/ Permeabilization buffer | BD Biosciences | 554714 |
| CellTrace™ Violet Cell Proliferation Kit | Invitrogen | C34557 |
| 7-AAD (7-Aminoactinomycin D) | Invitrogen | A1310 |
| LIVE/DEAD Fixable Yellow Dead Cell Stain Kit | Invitrogen | L34967 |
| HiSpeed Plasmid Midi Kit | Qiagen | 12643 |
| CRISPR/Cas9 knock-out kit | Takara Lenti X | 632629 |
| GSK2656157 | Cayman | 17372 |
| SB203580 | Cayman | 13067 |
| W146 | Tocris | 3602 |
| ATP assay kit | Sigma | 213-579-1 |
| CHIP Kit | Cell Signaling Technology | 9003 |
| S1P | Sigma | S9666 |
| Lonza electroporation buffer P3 kit | Lonza | Cat no. V4XP-3032 |

| Reagent/resource | Reference or source | Identifier or catalog number |
|---|---|---|
| **Software/algorithm** | **Source** | **Link** |
| FlowJo (v10) | BD Biosciences | https://www.flowjo.com/ |
| GraphPad Prism (v9) | GraphPad Software | https://www.graphpad.com/scientific-software/prism/ |
| Wave Software (Seahorse Analysis) | Agilent Technologies | https://www.agilent.com/ |
| ImageJ | NIH | https://imagej.net/ij |
| R studio | R studio | Posit \| The Open-Source Data Science Company |
| **Others** | | |
| BD LSR Fortessa II Flow Cytometer | BD Biosciences | |
| Lonza Nucleofector 4D | Lonza | |
| Confocal Microscope | Zeiss | |
| Bio-Rad CFX96 Real-Time PCR System | Bio-Rad | |
| Multiskan SkyHigh Microplate Spectrophotometer | Thermo Fisher Scientific | |
| Bio-Rad ChemiDoc Imaging System | Bio-Rad | |
| Agilent Seahorse XFe24 Analyzer | Agilent Technologies | |
| Eppendorf Centrifuge 5810 R | Eppendorf | |
| T100 Thermal Cycler | Bio-Rad | |

## Mice

C57BL/6 mice (JAX stock #000664) and Pmel-1 mice (JAX stock #005023) were procured from Jackson Laboratories, USA, and were housed under specific pathogen-free (SPF) conditions in the CSIR-IICB animal facility with controlled temperature, humidity, and a 12-h light/dark cycle, and were provided autoclaved food and water ad libitum under the supervision of trained veterinary staff. All procedures complied with institutional and national ethical guidelines and were approved by the CSIR-IICB Institutional Animal Ethics Committee (Ref No. IICB/AEC/Meeting/July/2020/4, dated 24/07/2020). Mice were routinely monitored for tumor burden, activity, and overall health, with humane endpoints applied when necessary. For tumor implantation and analysis, 8-10-week-old mice of both sexes were randomly assigned to experimental groups; no sex-specific differences were observed, and data from males and females were therefore pooled for final analyses.

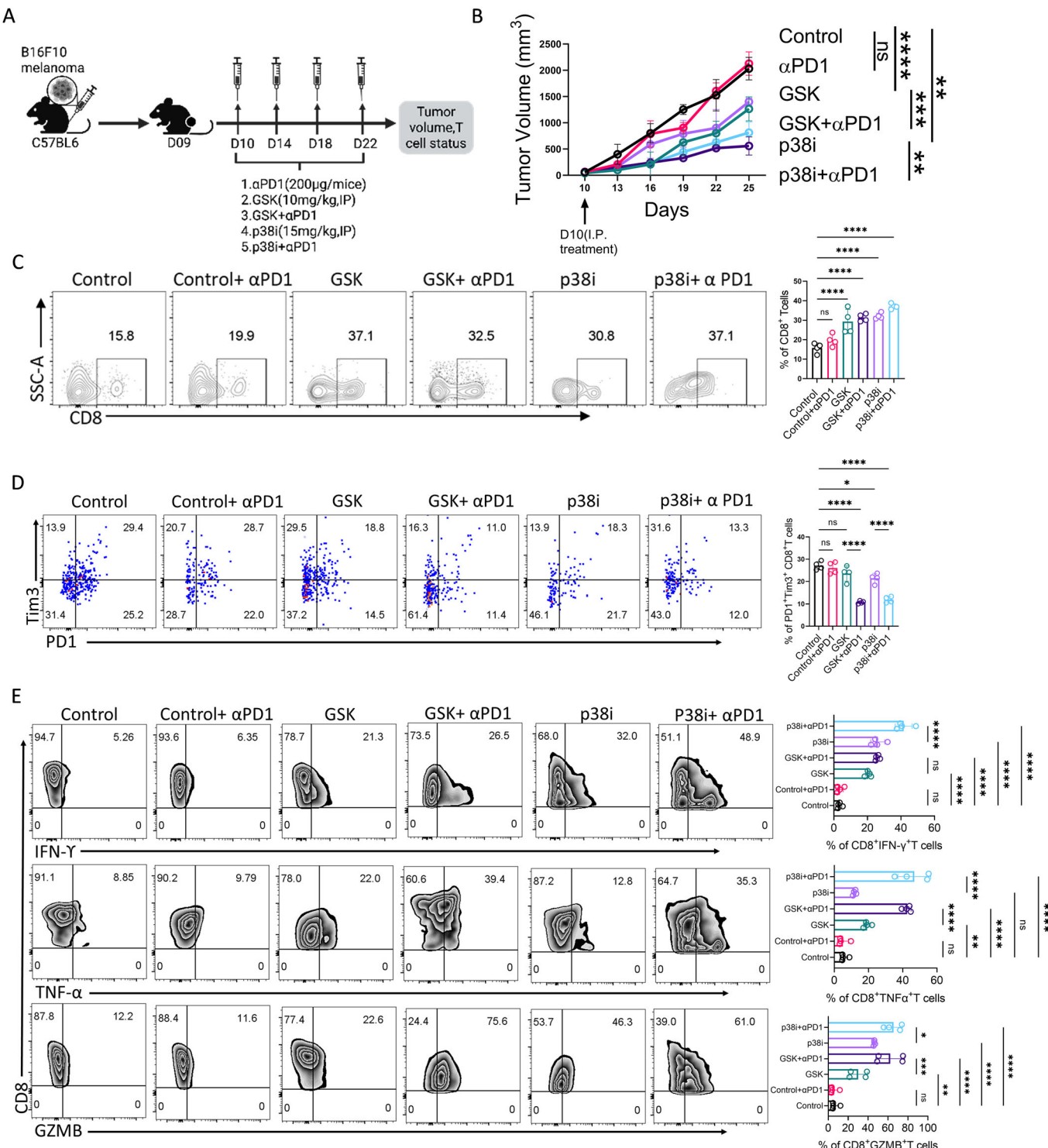

**Figure 7. PERK-CHOP-p38MAPK inhibition overcomes anti-PD-1 therapy resistance.**

(A–E) C57BL/6 mice ($n = 4$ mice/group) bearing subcutaneous B16F10 melanoma tumor were treated with either p38i, GSK, or vehicle control, alone or in combination with anti-PD1 antibody (Clone# RMP1-14; 200 μg/mouse twice weekly), as schematically shown in (A), and were evaluated for: (B) tumor growth, (C) frequency of CD8+ T cells at the tumor site, (D) frequency of terminally exhausted CD8+ T cells (PD1+Tim3+) at the tumor site, and (E) the ability of CD8+ T cells from the tumor site to produce different effector cytokines. The adjacent bar plot represents cumulative data from four tumor-bearing mice. *P < 0.05; **P < 0.01; ***P < 0.005; ****P < 0.0001; ns, nonsignificant (P > 0.05), the error bar represents the standard deviation (SD). P values are derived from unpaired one-way ANOVA (C–E), and two-way ANOVA test (B). Source data are available online for this figure.

## Cell lines

EL4 (TIB-39, ATCC), B16-F10 (CRL-6475, ATCC), and YUMM1.7 (CRL-3362, ATCC) tumor cell lines were obtained from the American Type Culture Collection (ATCC, USA) and maintained as authenticated, Mycoplasma-free stocks. Upon receipt, cells were revived, expanded for two passages, and cryopreserved as master stocks. Routine Mycoplasma testing using the MycoAlert Mycoplasma Detection Kit (Lonza) ensured only contamination-free cultures were used. Throughout the study, cells between passages 4 and 6 were used for all in vitro assays and tumor implantation experiments to maintain consistency and minimize culture-induced variation. Cells were grown in complete DMEM with 10% FBS, L-glutamine, and penicillin–streptomycin, and maintained at 37 °C in a humidified 5% $CO_2$ incubator with regular monitoring for morphology, growth kinetics, and viability.

## Isolation of tumor-infiltrating T cells

- Tumors were harvested aseptically from B16-F10, EL-4, and YUMM1.7 tumor–bearing mice and immediately transferred to cold RPMI-1640 medium. Excised tumors were finely minced using sterile scalpels.
- Minced tissue was enzymatically digested in RPMI-1640 containing:
  - Collagenase type IV (2 mg/mL)
  - DNase I (100 μg/mL)
- Digestion was performed at 37 °C for 45 min with gentle agitation.
- The digested tissue was passed through a 70 μm cell strainer to obtain a single-cell suspension.
- Red blood cells were lysed using ACK lysis buffer, when required.
- For enrichment of tumor-infiltrating lymphocytes (TILs), the cell suspension was layered over Hi-Sep LSM (HiMedia Laboratories, India).
- Density gradient centrifugation was performed at 1200 rpm for 30 min at room temperature.
- Mononuclear cells at the interface were carefully collected.
- Cells were washed and resuspended in RPMI-1640 for downstream analyses.

## T cell culture condition

Purified naive T cells or total splenocytes from B6 mice were activated with plate-bound anti-CD3 (5 mg/ml) and anti-CD28 (2 mg/ml) in the presence of rIL2 (100 U/ml) for 3 days. In some cases, T cells were activated either in the presence of vehicle control or GSK 2656157 (5 μm), SB203503 (10 μm), S1P (5 μM), after 24 h of T cell activation. Complete RMPI-1640 media supplemented with 10% FCS, 4 mmol/L L-glutamine, 100 U/mL penicillin, 100 mg/ml streptomycin, 55 mmol/L beta-mercaptoethanol was used for T-cell culture. For evaluation of intracellular cytokines, T cells were re-stimulated with PMA (500 ng/ml) and Ionomycin (1000 ng/ml) for 4 h in the presence of Golgi inhibitors and subsequently used for staining.

## Chronic stimulation of human and murine CD8+ T cells

- Peripheral blood mononuclear cells (PBMCs) were isolated from buffy coats of healthy donors using Ficoll–Hypaque density gradient centrifugation.
- Buffy coats were de-identified prior to use, and all procedures were approved by the Institutional IRB (No. IICB/IRB/2020/2 P).
- CD8+ T cells were purified from PBMCs using magnetic bead–based separation.
- Purified 1*$10^6$ CD8+ T cells were activated for 72 h with plate-bound anti-CD3 (5 μg/mL) and anti-CD28 (2 μg/mL) in the presence of recombinant human IL-2 (100 U/mL).
- Following activation, cells were cultured for an additional 12 days under one of the following conditions:
  - Continuous stimulation with plate-bound anti-CD3 (5 μg/mL), or
  - Culture with IL-2 alone (100 U/mL).
- Data were collected at the end of the 12-day culture period.
- For the mouse in vitro chronic exhaustion model:
  - $1 \times 10^6$ CD8+ T cells were activated for 2 days.
  - Cells were then subjected to continuous stimulation with plate-bound mouse anti-CD3 (5 μg/mL) for an additional 6 days. Cultures were split every 2 days during the continuous stimulation period.

## Flow cytometry staining

For cell surface staining, cultured cells were washed with FACS buffer (PBS with 0.1% BSA) and incubated with fluorochrome-conjugated antibodies (1:200) for 30 min at 4 °C in the dark, followed by washing to remove unbound antibodies. Intracellular cytokine staining was performed after stimulation with PMA (50 ng/mL) and ionomycin (500 ng/mL) in the presence of Brefeldin A for 4 h at 37 °C, after which cells were fixed and permeabilized using the BD Cytofix/Cytoperm Kit (BD Biosciences, USA) and stained with cytokine-specific antibodies. For transcription factor staining, cells were processed using the Foxp3/ Transcription Factor Staining Buffer Set (Thermo Fisher Scientific, USA) and incubated with the appropriate fluorochrome-labeled antibodies. All samples were washed with Perm/Wash buffer prior to acquisition on an LSR Fortessa flow cytometer, using compensation and fluorescence-minus-one (FMO) controls to ensure accurate gating. Data were analyzed with FlowJo software through sequential gating to remove doublets and dead cells and to quantify T-cell subsets, cytokine-producing populations, mean fluorescence intensity, and other functional readouts.

## Apoptotic assay

- To assess activation-induced cell death (AICD), previously activated T cells were restimulated overnight with plate-bound anti-CD3 at 37 °C.
- The following day, cells were washed with cold FACS buffer (PBS containing 0.1% BSA).
- Cells were stained with fluorochrome-conjugated surface antibodies (1:200 dilution) for 30 min at 4 °C.
- After surface staining, cells were washed thoroughly to remove excess antibodies.

- Apoptosis was assessed using Annexin V and 7-AAD staining.
- Cells were resuspended in Annexin binding buffer (BD Biosciences, USA).
- Cells were incubated with Annexin V for 20 min at room temperature.
- 7-AAD was added immediately before data acquisition.

Samples were acquired on an LSR Fortessa flow cytometer. Flow cytometry data were analyzed using FlowJo software with standard gating strategies, including exclusion of doublets and quantification of apoptotic subsets.

## ATP assay

- Activated CD8$^+$ T cells ($1 \times 10^6$ cells per condition) were used for intracellular ATP quantification. ATP levels were measured using the ATP Bioluminescent Assay Kit (Sigma-Aldrich, USA) according to the manufacturer's instructions.
- T cells were harvested post-activation and washed twice with ice-cold PBS. Cells were lysed to release intracellular metabolites.
- Lysates were centrifuged at high speed to remove cellular debris.
- Clarified supernatants were transferred to pre-chilled tubes.
- Aliquots of each supernatant were mixed with the luciferase-based detection reagent.
- Reactions were performed in white opaque 96-well plates to minimize background and enhance signal detection.
- Luminescence was measured immediately after reagent addition using a Varioskan LUX Multimode Reader (Thermo Fisher Scientific, USA) with optimized integration settings.

## Immunoblotting

To evaluate the expression levels of CHOP, phosphorylated p38 (p-p38), and total p38, cells were lysed utilizing RIPA buffer supplemented with a protease inhibitor cocktail (Thermo Fisher Scientific). Proteins were standardized to equal amounts (30 µg) and resolved through 10% SDS-polyacrylamide gel electrophoresis (SDS-PAGE) before being transferred onto nitrocellulose membranes for further analysis. The membranes were incubated overnight at 4 °C with primary antibodies against CHOP and p-p38 (Invitrogen), followed by treatment with secondary antibodies conjugated to horseradish peroxidase (HRP)—specifically goat anti-mouse IgG diluted at 1:10,000 (Jackson Immuno Research Laboratories)—for two hours at ambient temperature. Prestained protein standards (Invitrogen) were run simultaneously to facilitate the determination of molecular weights. Chemiluminescence-based detection was carried out using Clarity Western ECL substrate (Bio-Rad), and signal intensities were captured with the Bio-Rad Versadoc Imaging System (Bio-Rad). Subsequently, the same membranes were stripped of bound antibodies using stripping buffer and re-incubated with an antibody targeting β-actin (Cell Signaling Technology) to serve as a normalization control.

## qPCR

Quantitative reverse transcription PCR (qPCR) was performed to assess gene expression in T cells. Total RNA was extracted using TRIzol reagent (Thermo Fisher Scientific, USA) following the manufacturer's protocol, including chloroform phase separation, isopropanol precipitation, ethanol washing, and resuspension in nuclease-free water. RNA quantity and purity were assessed using a Multiscan SkyHigh Plate Reader (Thermo Fisher Scientific, USA). For cDNA synthesis, 1 µg of RNA was reverse-transcribed using the iScript cDNA Synthesis Kit (Bio-Rad, USA), and the resulting cDNA was diluted 1:3 for qPCR. Reactions were set up in triplicate using iTaq Universal SYBR Green Supermix (Bio-Rad, USA) with 250 nM gene-specific primers in 96-well plates and run on a CFX96 Real-Time PCR System (Bio-Rad, USA). Cycling conditions included 95 °C for 3 min, followed by 40 cycles of 95 °C for 10 s and 60 °C for 30 s, with melt-curve analysis to confirm specificity. Ct values were analyzed using the ΔΔCt method, normalizing to Actb, and results were expressed as fold change relative to controls.

Details of all primer sequences utilized are provided in the Reagents and Tools table.

## Metabolic flux analysis

Extracellular acidification rate (ECAR) and oxygen consumption rate (OCR) were measured using the Seahorse XFe24 Analyzer (Agilent Technologies, USA) to evaluate glycolytic activity and mitochondrial respiration in purified CD8$^+$ T cells. Cells ($0.5 \times 10^6$ per well) were seeded onto Cell-Tak-coated XF24 microplates and allowed to adhere for 30 min in a non-$CO_2$ incubator. ECAR was assessed using the glycolysis stress test, with sequential injections of glucose (10 mmol/L), oligomycin (1 mmol/L), and 2-deoxy-D-glucose (100 mmol/L) to determine glycolysis, glycolytic capacity, and glycolytic reserve. OCR was measured using the mitochondrial stress test, with stepwise injections of oligomycin (1 mmol/L), FCCP (1 mmol/L), and rotenone/antimycin A (2 µmol/L + 100 nmol/L) to quantify ATP-linked respiration, maximal respiration, and non-mitochondrial oxygen consumption.

## siRNA mediated knockdown

- CD8$^+$ T cells were purified from splenocytes of C57BL/6 mice. Cells were activated for 48 h using plate-bound anti-CD3 and anti-CD28 antibodies.
- Activated CD8$^+$ T cells ($2 \times 10^6$ cells per reaction) were washed with serum-free medium prior to transfection.
- Cells were transfected with either S1pr1-specific siRNA or non-targeting control siRNA using the Lonza Nucleofector Kit (V4XP-3024) and the manufacturer-recommended program for primary murine T cells (DN 100).
- Immediately after electroporation, cells were transferred into pre-warmed complete RPMI medium supplemented with IL-2.
- Cells were cultured for an additional 48 h to allow efficient S1pr1 knockdown.
- S1P (5 µM) was added at 24 h post-transfection to both control groups and S1pr1 knockdown groups. Post-transfection, cells were divided into four experimental groups:
  - Control siRNA
  - Control siRNA + S1P
  - S1pr1 siRNA
  - S1pr1 siRNA + S1P.

For protein extraction, cells were washed with ice-cold PBS and lysed in RIPA buffer containing protease inhibitors. Lysates were incubated on ice for 30 min, briefly sonicated, and centrifuged at 12,000×g for 15 min at 4 °C. Equal amounts of protein (30 μg per lane) were mixed with reducing Laemmli buffer and boiled for 5 min. Proteins were separated by SDS-PAGE and analyzed by immunoblotting to confirm S1pr1 knockdown and assess downstream signaling changes.

## Adoptive T cell transfer

- Wild-type (WT) B6 mice (6–8 weeks old) were used for all experiments.
- Mice were subcutaneously implanted with B16-F10 melanoma cells ($0.5 \times 10^6$ cells/mouse).
- On day 8 post-tumor implantation, WT B6 mice were lymphodepleted using cyclophosphamide (4 mg/mouse).
- Activated Pmel T cells were prepared by stimulating cells for 48 h with gp100 peptide (3 μg/mL).
- Pmel T cells were transduced with control shRNA, shS1pr1, or shEif2ak3.
- Activated and transduced Pmel T cells ($1.5 \times 10^6$ cells/mouse) were adoptively transferred intravenously.
- Recombinant IL-2 (50,000 U/mouse) was administered intraperitoneally for 3 consecutive days following adoptive transfer.
- In selected experiments, mice received anti-PD-1 antibody (200 μg/mouse) or isotype control IgG, administered three times per week.

## shRNA mediated knockdown of T cells

- HEK 293 T (ATCC) cells were cultured in T75 tissue culture flasks (Falcon, NY, USA) under standard conditions (37 °C, 5% $CO_2$). Cells were maintained in complete DMEM (Gibco, Thermo Fisher Scientific) supplemented with:10% fetal bovine serum (FBS; Gibco), 1% penicillin–streptomycin (Gibco)
- For lentiviral production, HEK293T cells were co-transfected with:
  - 15 μg lentiviral expression plasmid encoding S1pr1 shRNA, Eif2ak3 shRNA, or non-targeting scrambled shRNA control (Origene, USA)
  - 10 μg psPAX2 packaging plasmid
  - 5 μg pMD2.G envelope plasmid
- Transfection was carried out using the $CaCl_2$/HBS precipitation method.
- At 24 h post-transfection, the culture medium was replaced with low-serum medium.
- Cells were incubated for an additional 48–72 h to allow viral production.
- Lentivirus-containing supernatant was harvested and clarified by centrifugation at 500×g for 5 min. The supernatant was filtered through a 0.45 μm polyethersulfone membrane filter (MilliporeSigma).
- Viral particles were concentrated using Lenti-X Concentrator (Takara Bio, Shiga, Japan) according to the manufacturer's instructions. Concentrated virus was aliquoted and stored at −80 °C until use.

- Activated $1 \times 10^6$ CD8⁺ T cells were washed twice with PBS. Cells were transferred to non-tissue culture-treated 24-well plates (Falcon) pre-coated overnight at 4 °C with RetroNectin (30 μg/mL; Takara Bio).
- CD8⁺ T cells were transduced with concentrated lentiviral particles.
- Spinoculation was performed at 1000×g for 1.5 h at 32 °C.
- Fresh IL-2 (100 U/mL) was added immediately after spinoculation. Cells were cultured for an additional 24 h at 37 °C in a humidified 5% $CO_2$ incubator prior to downstream analyses.
- Following spinoculation, sphingosine-1-phosphate (S1P) was added in the designated concentration to stimulate signaling pathways of interest. The transduced cells were then incubated under physiological conditions at 37 °C for an additional 48 h, after which they were harvested and processed for downstream experimental analyses.

## CRISPR/Cas9-mediated knockdown

- sgRNA targeting Sphk1 was designed using the CHOPCHOP online tool.
- Complementary oligonucleotides corresponding to the sgRNA were annealed to generate a double-stranded DNA duplex.
- The annealed sgRNA duplex was cloned into the linearized pLVX-hyg-sgRNA1 vector.
- Lentiviral particles were produced by transfecting Lenti-X Packaging Single Shots with Cas9 and sgRNA plasmids.
- Following a 10-min incubation, the transfection mixture was applied to HEK 293T cells.
- At 24 h post-transfection, the culture medium was replaced with a low-serum formulation to enhance viral production.
- Cells were maintained for an additional 72 h to allow lentiviral amplification.
- Virus-containing supernatant was harvested, filtered to remove cellular debris, and concentrated using Lenti-X Concentrator reagent (Takara Bio) according to the manufacturer's instructions.
- YUMM1.7 cells were transduced with LVX-hyg-sgRNA1 lentivirus in the presence of polybrene.
- Transduction efficiency was enhanced by centrifugation for 30 min.
- Hygromycin selection was applied to generate a stable sgRNA-expressing cell population.
- Hygromycin-resistant cells were subsequently transduced with LVX-puro-Cas9 lentivirus under identical conditions.
- Puromycin selection was performed to enrich for Cas9-expressing cells.
- Dual antibiotic–resistant cells were serially diluted into 96-well plates and cultured under hygromycin and puromycin selection.
- A single-cell clone was isolated and expanded for two weeks to establish Sphk1 knockout YUMM1.7 cells.

## Integrated TFBS prediction and ChIP assay

The 1000 bp upstream promoter sequences of the *MAP3K8, Map3k13, Map3k15* gene (mouse, GRCm39 assembly) were retrieved using the Ensembl Genome Browser (https://asia.ensembl.org/index.html) (Martin et al, 2023). Transcription factor binding site (TFBS) prediction was performed via the TFBIND

webserver (https://tfbind.hgc.jp/) (Tsunoda and Takagi, 1999), which uses TRANSFAC-derived weight matrices. Analysis revealed multiple high-confidence CHOP (C/EBP homologous protein) binding sites within the promoter region. The position frequency matrix for CHOP was obtained from the JASPAR database (https://jaspar.elixir.no/) (Rauluseviciute et al, 2024).

- Chromatin immunoprecipitation (ChIP) assays were performed using murine CD8$^+$ T cells stimulated in vitro with sphingosine-1-phosphate (S1P).
- Approximately $1 \times 10^7$ S1P-activated cells were fixed with 1% paraformaldehyde for 10 min at room temperature to crosslink protein–DNA complexes.
- Crosslinking was quenched with 125 mM glycine.
- Nuclei were isolated using buffer A containing DTT.
- Chromatin was digested with micrococcal nuclease and briefly sonicated to generate DNA fragments of ~200–500 bp.
- Chromatin was diluted in 1× ChIP buffer.
- Samples were incubated overnight at 4 °C with:
  - Anti-CHOP antibody (2 μg; Cat. #9649S, Cell Signaling Technology), or
  - Control rabbit IgG.
- Immune complexes were captured using Protein G magnetic beads.
- Beads were washed and chromatin was eluted at 65 °C.
- Crosslinks were reversed using 5 M NaCl.
- Proteins were digested with Proteinase K.
- DNA was purified using spin columns and eluted in the supplied elution buffer.
- Enriched DNA was quantified by SYBR Green–based qPCR using primers targeting predicted CHOP-binding sites in the Map3K8, Map3k13, and Map3k15 promoters.
- Melt curve analysis confirmed amplicon specificity.
- Relative enrichment was calculated using the percent input method, normalized to 2% input DNA.

Percent Input = 2% x $2^{(C[T]\ 2\%Input\ Sample\ -\ C[T]\ IP\ Sample)}$

C[T] = $C_T$ = Threshold cycle of PCR reaction.

### RNA sequencing and analysis

Total RNA was extracted from six cell pellets ($1.5 \times 10^6$ cells each) suspended in 1 mL TRIzol (Invitrogen) and transported on dry ice to LC GC Laboratories. RNA was purified using the RNeasy Mini Kit (Qiagen, Cat. No. 74106), and quantified with both NanoDrop ND-100 and Qubit 4 Fluorometer. RNA integrity was assessed via Agilent Tapestation 4200.

For library preparation, 500 ng of RNA per sample was processed using the KAPA mRNA Capture Kit (Roche) and the KAPA RNA HyperPrep Kit for Illumina. mRNA was enriched using oligo-dT beads, fragmented, and reverse transcribed to cDNA using random hexamers. Strand-specific libraries were constructed with dUTP incorporation and adapter ligation, followed by PCR amplification.

Sequencing was performed on an Illumina NextSeq 2000 platform, generating ~40 million 2×150 bp paired-end reads per sample. Over 90% of reads had Q30 scores above 75%. FASTQ files were quality-checked using FastQC and trimmed with Trim Galore. Cleaned reads were aligned to the mouse genome using STAR (Dobin et al, 2013), and gene-level counts were generated with

featureCounts (Liao et al, 2014). Differential expression analysis was performed using DESeq2 (Love et al, 2014) in R. Functional enrichment and visualization were carried out using gprofiler2, rrvgo, made4, and EnhancedVolcano R packages.

### Statistical analysis

All data reported are the arithmetic mean from three or five biological replicates experiments performed in triplicate ±SD unless stated otherwise. Comparisons between two groups were performed using an unpaired, two-tailed, Student $t$ test. Comparisons between more than two groups were performed using one-way or two-way ANOVA. Data analyses were performed using the GraphPad Prism 9 software (GraphPad).

## Data availability

The RNA-sequencing data have been deposited in the NCBI Gene Expression Omnibus (GEO) repository and are publicly available under the accession number GSE313941. The raw FASTQ files are available through NCBI under BioProject accession PRJNA1380112.

The source data of this paper are collected in the following database record: biostudies:S-SCDT-10_1038-S44319-026-00734-3.

## Peer review information

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

## Acknowledgements

This work was supported by the SERB, India Grant CRG/2019/001334. SCha
also acknowledges funding from MLP-121 and P50 from CSIR-IICB, Kolkata, and
support from the Central Instrumentation Facility (CIF) of CSIR-IICB, Kolkata.
A.G. acknowledges support from the High Performance and Cloud Computing
Group at the Zentrum für Datenverarbeitung of the University of Tübingen, the
state of Baden-Württemberg through bwHPC, and the German Research
Foundation (DFG) through grant no. INST 37/935-1 FUGG. AG also receives
support from the de.NBI Cloud within the German Network for Bioinformatics
Infrastructure (de.NBI) and ELIXIR-DE (Forschungszentrum Jülich and
W-de.NBI-001, W-de.NBI-004, W-de.NBI-008, W-de.NBI-010, W-de.NBI-013,
W-de.NBI-014, W-de.NBI-016, W-de.NBI-022) to conduct computational
analysis in this work.

## Author contributions

**Debashree Basak**: Conceptualization; Formal analysis; Investigation;
Methodology; Writing—original draft; Writing—review and editing. **Puspendu
Ghosh**: Formal analysis; Investigation; Methodology. **Anupam Gautam**: Formal
analysis; Methodology. **Ishita Sarkar**: Investigation. **Arpita Bhoumik**:
Investigation. **Soham Chowdhury**: Investigation. **Shaun Mahanti**: Investigation.
**Anwesha Mandal**: Methodology. **Rajeswari Chakraborty**: Methodology.
**Anwesha Kar**: Investigation. **Snehanshu Chowdhury**: Investigation. **Krishna
Kumar**: Methodology. **Shubhrajit Barman**: Methodology. **Senthil Kumar
Ganesan**: Methodology. **Saikat Chakrabarti**: Methodology. **Sandip Paul**: Formal
analysis; Methodology. **Shilpak Chatterjee**: Conceptualization; Resources;
Supervision; Funding acquisition; Writing—original draft; Project
administration; Writing—review and editing.

Source data underlying figure panels in this paper may have individual
authorship assigned. Where available, figure panel/source data authorship is
listed in the following database record: biostudies:S-SCDT-10_1038-S44319-
026-00734-3.

## Disclosure and competing interests statement

The authors declare no competing interests.

# Expanded View Figures

**Figure EV1. S1P-S1PR1 signaling modulates exhaustion profiles of CD8⁺ T cells in TME and chronic stimulation models.** ▶

(A–C) Gating strategy used for flow cytometry analysis of heterogeneous exhaustion states of CD8⁺ T cells within the tumor microenvironment (TME) of (A) EL4 thymoma ($n = 4$), (B) B16F10 melanoma ($n = 5$), (C) YUMM1.7 melanoma ($n = 7$), distinguished by the differential expression of PD1 and Tim3. (D–H) Human CD8⁺ T cells isolated from healthy donor PBMCs were activated for 3 days and then subjected to either continuous TCR stimulation or cultured (without TCR stimulation) with IL-2 for 15 days. Cells were analyzed for (D) exhaustion-associated surface markers, (E) intracellular production of IFNγ and TNFα, (F) intracellular expression of transcription factor TCF1, (G) intracellular expression of cell proliferation marker Ki67, and (H) cell death using Annexin V and 7AAD. The adjacent bar diagram represents cumulative data from $n = 4$ biological replicates. (I–M) CD8⁺ T cells isolated from the spleen of wild-type B6 mice were activated for 2 days and then subjected to either continuous TCR stimulation or cultured (without TCR stimulation) with IL-2 culture for 9 days. Cells were assessed for (I) exhaustion-associated surface markers, (J) intracellular production of IFNγ and TNFα, (K) intracellular expression of the transcription factor TCF1, (L) intracellular expression of cell proliferation marker Ki67, and (M) cell death using Annexin V and 7AAD. The adjacent bar diagram represents cumulative data from $n = 4$ biological replicates. (N, O) Flow cytometry analysis of S1PR1 expression in acutely versus chronically stimulated human (N) and murine (O) CD8⁺ T cells. The adjacent bar diagram represents cumulative data from $n = 4$ biological replicates. (P) Flow cytometry analysis of p-STAT3 expression in CD8⁺ T cells isolated from spleens versus tumor tissues, with adjacent bar graphs representing cumulative results from four biological replicates ($n = 4$). (Q) Purified mouse CD8⁺ T cells activated in the presence or absence of S1P were assessed for flow cytometry-based expression of CD25. The adjacent bar diagram represents cumulative data from three biological replicates ($n = 3$). $*P < 0.05$; $**P < 0.01$; $***P < 0.005$; $****P < 0.0001$; ns, nonsignificant ($P > 0.05$), the error bar represents the standard deviation (SD). $P$ values are derived from unpaired two-tailed Student's $t$ test (A–P). Source data are available online for this figure.

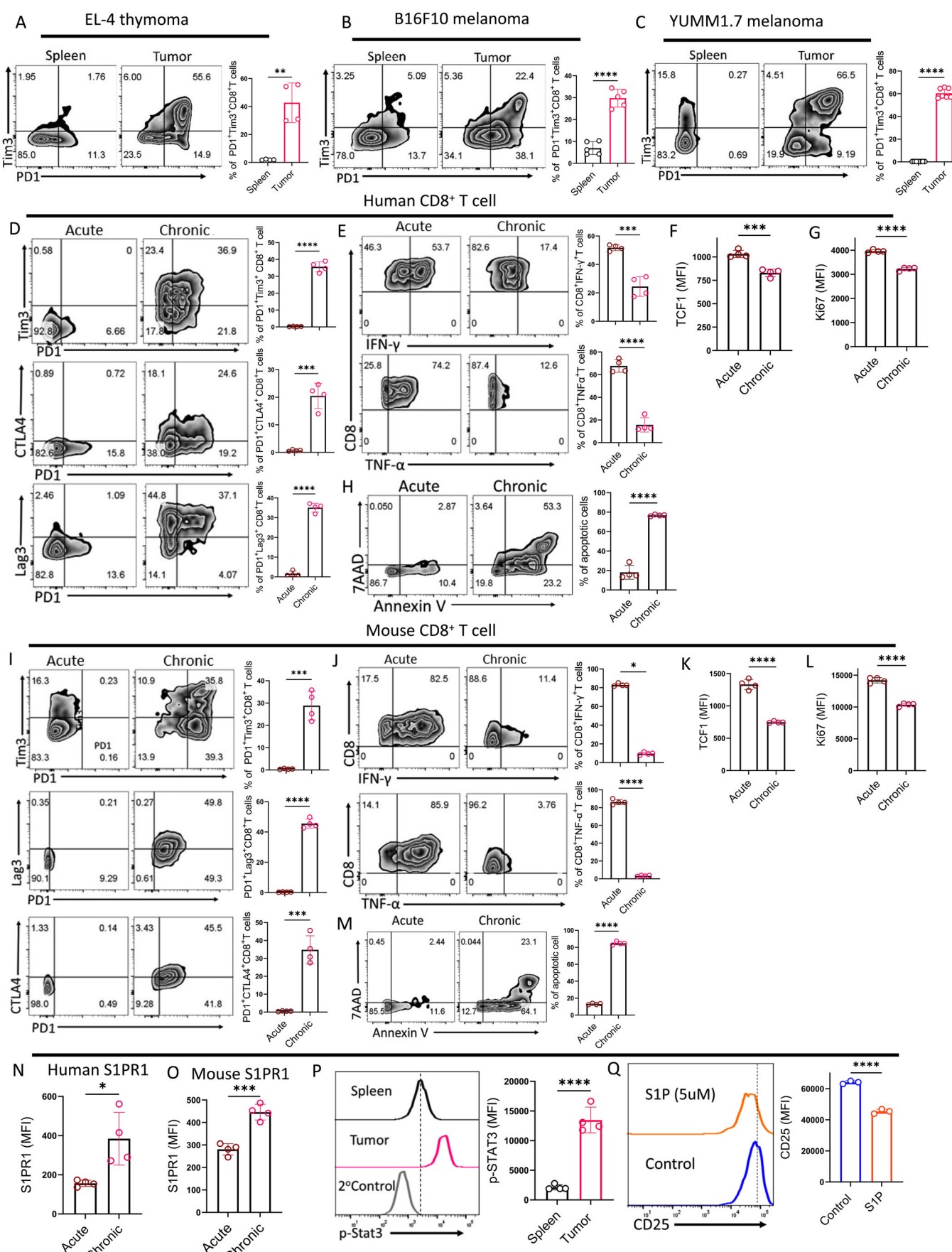

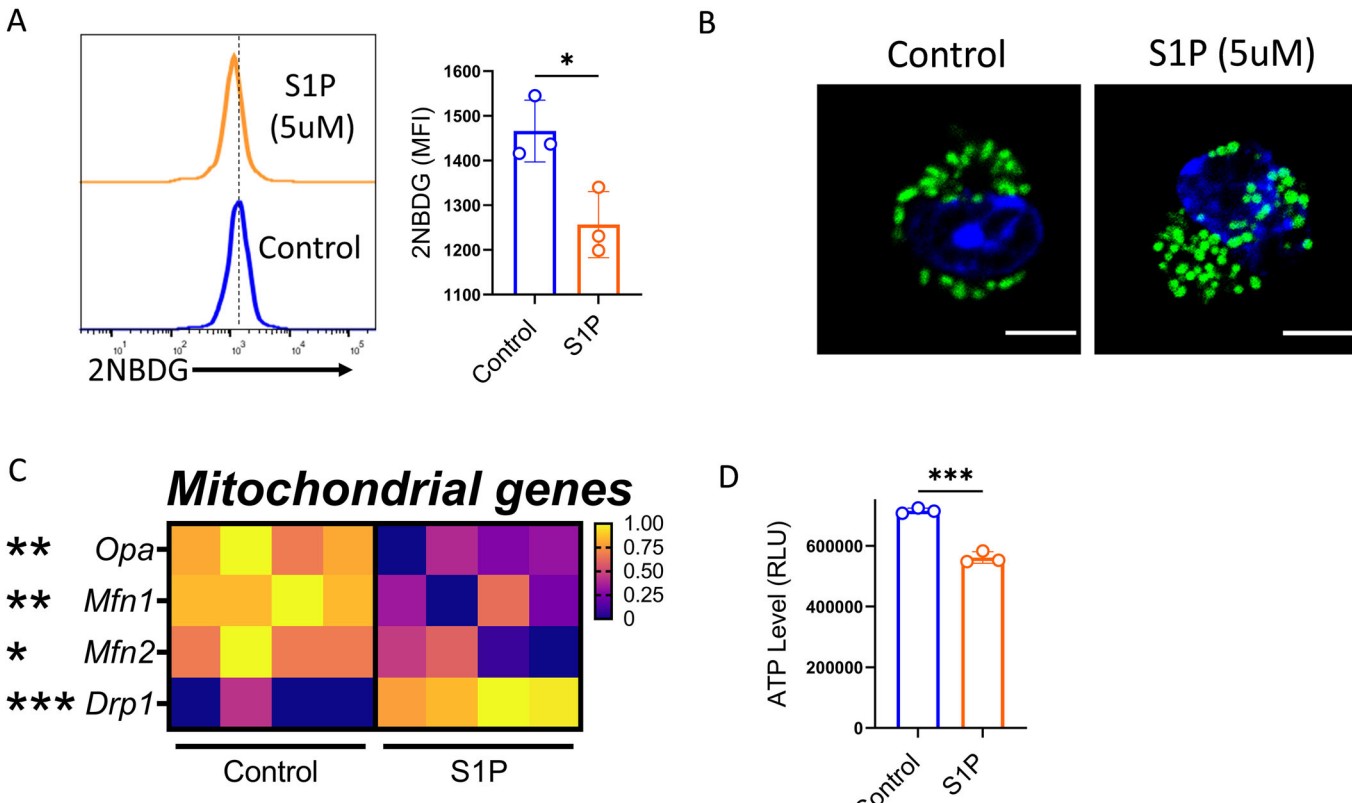

**Figure EV2. S1P-S1PR1 signaling modulates metabolic profiles of CD8+ T cells.**

(A) Purified mouse CD8+ T cells activated in the presence or absence of S1P were assessed for glucose uptake using 2NBDG ($n = 3$). (B) Confocal imaging of activated CD8+T cells in the presence and absence of S1P. (C) qPCR analysis showing transcript levels of different genes involved in the mitochondrial pathway in respective groups ($n = 4$). (D) Graphical representation of the intracellular ATP level in S1P or vehicle-treated CD8+ T cells ($n = 3$). *$P < 0.05$; **$P < 0.01$; ***$P < 0.005$; ****$P < 0.0001$; ns, nonsignificant ($P > 0.05$), the error bar represents the standard deviation (SD). $P$ values are derived from unpaired two-tailed Student's $t$ test (A, D), and two-way ANOVA test (C). Source data are available online for this figure.

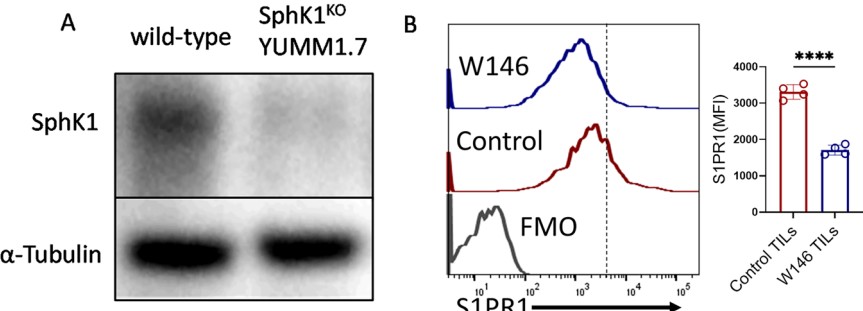

**Figure EV3.  SphK1 deletion in tumor cells and local S1PR1 inhibition reduce S1PR1 level in tumor-infiltrating T cells.**

(A) Western blot analysis showing expression of SphK1 in wild-type and SphK1$^{KO}$ YUMM1.7 cells ($n = 3$). (B) Flow cytometry analysis of the surface expression of S1PR1 on intratumoral CD8$^+$ T cells following administration of S1PR1 antagonist W146 or vehicle control in YUMM1.7 tumor-bearing mice ($n = 4$). *$P < 0.05$; **$P < 0.01$; ***$P < 0.005$; ****$P < 0.0001$; ns, nonsignificant ($P > 0.05$), the error bar represents the standard deviation (SD). $P$ values are derived from unpaired two-tailed Student's $t$ test (B). Source data are available online for this figure.

                                                                          

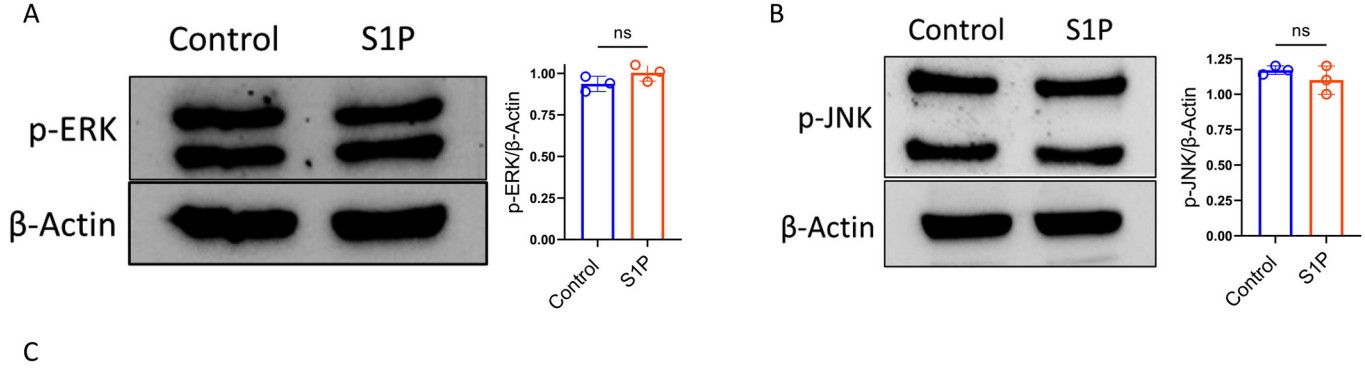

C

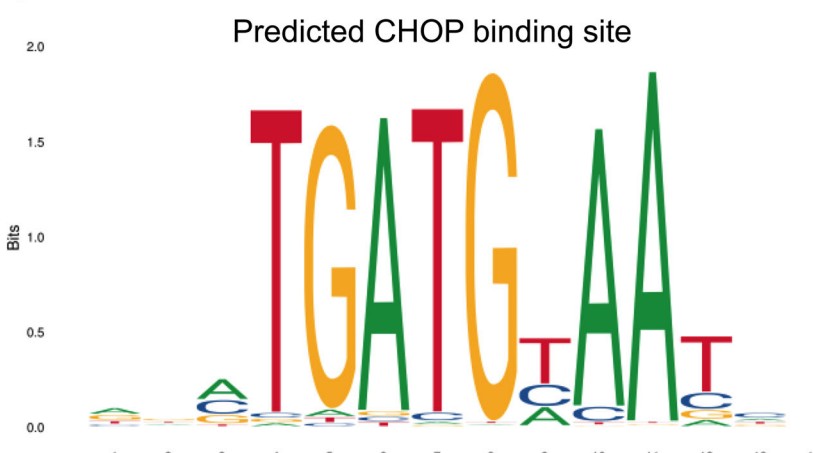

*Map3k8*   : +107  AAAAGCAAACCCC +119; Predicted binding score – 0.77
*Map3k13*  : +949  TGCTGCAATCTTC  +961; Predicted binding score – 0.86
*Map3k15*  :  -18  CTCTGAAATAACC  -6   ; Predicted binding score – 0.83

**Figure EV4.  Evaluation of different MAPK signaling by S1P-S1PR1 signaling.**

(**A**, **B**) Western blot analysis showing the expression of (**A**) p-ERK and (**B**) p-JNK in in-vitro activated CD8[+] T cells either in the presence or absence of S1P. The adjacent bar graph depicts normalized densitometric data from three biological replicates ($n = 3$). (**C**) Predicted binding sites of CHOP on the *Map3k8*, *Map3k13*, and *Map3k15* promoters generated from JASPER Software. $*P < 0.05$; $**P < 0.01$; $***P < 0.005$; $****P < 0.0001$ ns, nonsignificant ($P > 0.05$), the error bar represents the standard deviation (SD). $P$ values are derived from unpaired two-tailed Student's $t$ test (**A**, **B**). Source data are available online for this figure.

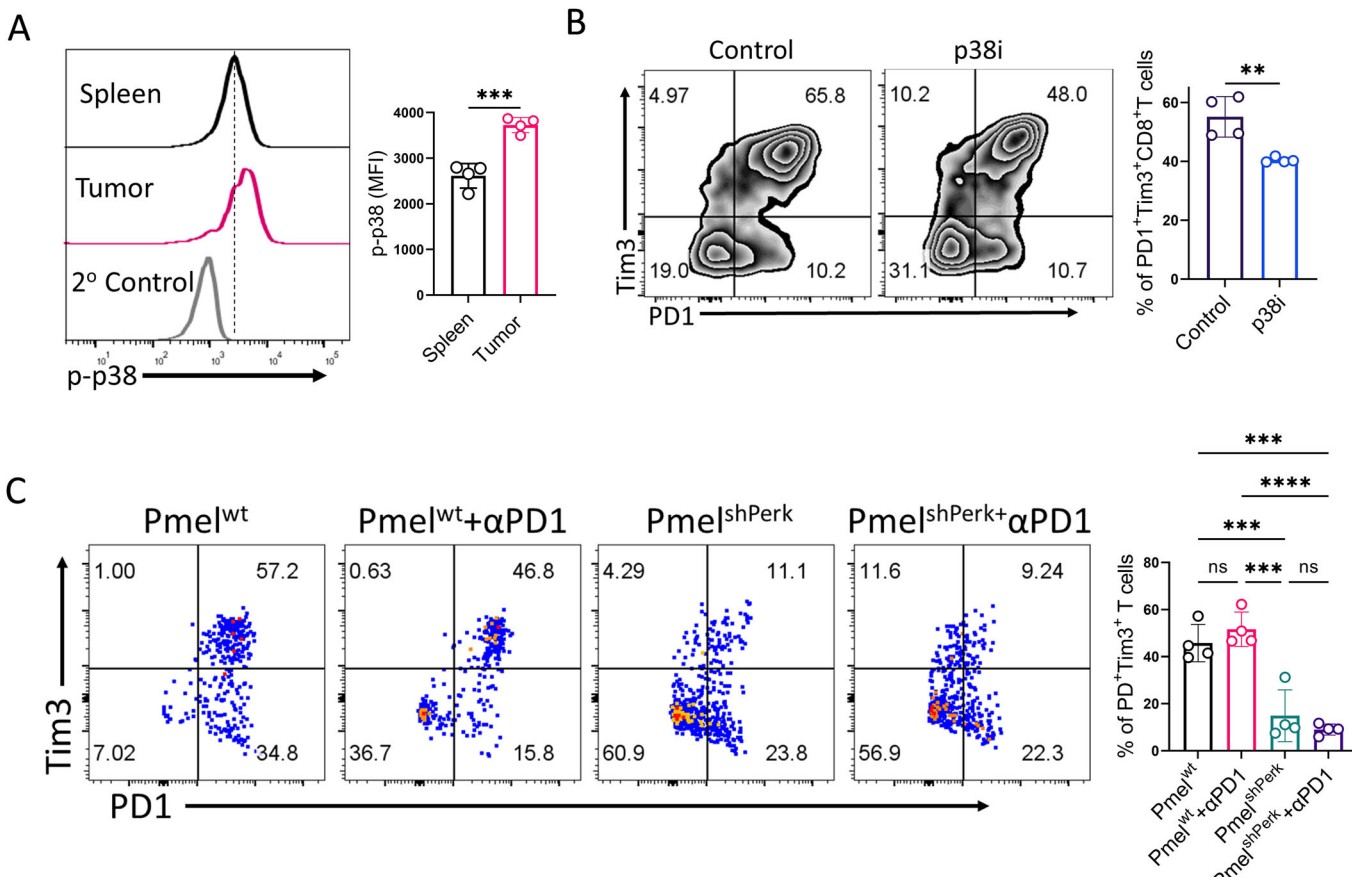

**Figure EV5. Evaluation of intratumoral CD8⁺ T cells.**

(A) CD8⁺ T cells isolated from either the tumor site or spleen of C57BL/6 mice ($n = 4$) bearing YUMM1.7 melanoma were assessed for p-p38 expression. The adjacent bar plot summarizes pooled data from four tumor-bearing mice. (B) Intratumoral CD8⁺ T cells from C57BL/6 mice ($n = 4$/group) with subcutaneous YUMM1.7 melanoma, treated with vehicle control or p38i, were evaluated for the frequency of terminally exhausted CD8⁺ T cells (PD1⁺Tim3⁺). The adjacent bar plot summarizes pooled data from four mice per group. (C) Adoptively transferred Pmel-1 T cells transduced with either control shRNA or shRNA targeting PERK, isolated from tumors of C57BL/6 mice ($n = 4$/group) bearing subcutaneous B16-F10 melanoma and treated with or without anti-PD1 antibody, were evaluated for the frequency of terminally exhausted CD8⁺ T cells (PD1⁺Tim3⁺). The adjacent bar plot summarizes pooled data from four mice per group. *$P < 0.05$; **$P < 0.01$; ***$P < 0.005$; ****$P < 0.0001$ ns, nonsignificant ($P > 0.05$). The error bar represents the standard deviation (SD). $P$ values are derived from unpaired two-tailed Student's $t$ test (A, B) and one-way ANOVA. Source data are available online for this figure.

