## [Peer Review File · EMBO Reports]

S1P-S1PR1 signaling impairs CD8+ T cell metabolism and effector function in tumors

Shilpak Chatterjee, Debashree Basak, Puspendu Ghosh, Anupam Gautam, Ishita Sarkar, Arpita Bhoumik, Soham Chowdhury, Shaun Mahanti, Anwesha Kar, Snehanshu Chowdhury, Krishna Kumar, Shubhrajit Barman, Senthil Ganesan, Saikat Chakrabarti, Sandip Paul, Anwesha Mandal, and Rajeswari Chakraborty

Corresponding author(s): Shilpak Chatterjee (schatterjee@iicb.res.in)

Review Timeline:

Submission Date:	7th Jul 25
Editorial Decision:	14th Aug 25
Revision Received:	12th Dec 25
Editorial Decision:	23rd Jan 26
Revision Received:	3rd Feb 26
Accepted:	19th Feb 26

Editor: Achim Breiling

Transaction Report:

Dear Dr. Chatterjee,

Thank you for the submission of your manuscript to EMBO reports. I have now received reports from the referees that were asked to evaluate your study, which can be found at the end of this email. As you will see, the referees think that these findings are of interest. However, they have several comments, concerns, and suggestions, indicating that a major revision of the manuscript is necessary to allow publication of the study in EMBO reports. As the reports are below, and all the referee concerns need to be addressed, I will not detail them here.

Given the constructive referee comments, I would thus like to invite you to revise your manuscript with the understanding that the concerns of the referees must be addressed in the revised manuscript and/or in a detailed point-by-point response. Acceptance of your manuscript will depend on a positive outcome of a second round of review. It is EMBO reports policy to allow a single round of revision only and acceptance of the manuscript will therefore depend on the completeness of your responses included in the next, final version of the manuscript.

1) a .docx formatted version of the final manuscript text (including legends for main figures, EV figures and tables), but without the figures included. Figure legends should be compiled at the end of the manuscript text.

2) individual production quality figure files as .eps, .tif, .jpg (one file per figure), of main figures and EV figures. Please upload these as separate, individual files upon re-submission.

4) a complete author checklist, which you can download from our author guidelines

(<https://www.embopress.org/page/journal/14693178/authorguide>). Please insert page numbers in the checklist to indicate where the requested information can be found in the manuscript. The completed author checklist will also be part of the RPF.

5) that primary datasets produced in this study (e.g. RNA-seq, ChIP-seq, structural and array data) are deposited in an appropriate public database. If no primary datasets have been deposited, please also state this in a dedicated section (e.g. 'No

primary datasets have been generated and deposited'), see below.

The accession numbers and database should be listed in a formal "Data Availability" section that follows the model below. This is now mandatory (like the COI statement). Please note that the Data Availability Section is restricted to new primary data that are part of this study. This section is mandatory. As indicated above, if no primary datasets have been deposited, please state this in this section

Data availability

6) We now request the publication of original source data with the aim of making primary data more accessible and transparent to the reader. You will receive a separate email with instructions for providing source data with your revised manuscript, including information how to upload and organize the files.

8) Regarding data quantification and statistics, please make sure that the number "n" for how many independent experiments were performed, their nature (biological versus technical replicates), the bars and error bars (e.g. SEM, SD) and the test used to calculate p-values is indicated in the respective figure legends (also for EV and Appendix figures). Please also check that all the p-values are explained in the legend, and that these fit to those shown in the figure. Please provide statistical testing where applicable. Please avoid the phrase 'independent experiment', but clearly state if these were biological or technical replicates. Please also indicate (e.g. with n.s.) if testing was performed, but the differences are not significant. In case n=2, please show the data as separate datapoints without error bars and statistics. See also: <http://www.embopress.org/page/journal/14693178/authorguide#statisticalanalysis>

9) Please add scale bars of similar style and thickness to all microscopic images, using clearly visible black or white bars (depending on the background). Please place these in the lower right corner of the images themselves. Please do not write on or near the bars in the image but define the size in the respective figure legend.

10) Please also note our reference format:

12) We now use CRedit to specify the contributions of each author in the journal submission system. CRedit replaces the author contribution section. Please use the free text box to provide more detailed descriptions and do NOT provide your final manuscript text file with an author contributions section. See also our guide to authors: <https://www.embopress.org/page/journal/14693178/authorguide#authorshipguidelines>

13) All Materials and Methods need to be described in the main text using our 'Structured Methods' format, which is required for all research articles. According to this format, the Methods section should include a Reagents and Tools Table (listing key

reagents, experimental models, software, and relevant equipment and including their sources and relevant identifiers), uploaded as separate file, and a Methods section in which we encourage the authors to describe their methods using a step-by-step protocol format with bullet points, to facilitate the adoption of the methodologies across labs. More information on how to adhere to this format as well as downloadable templates (.doc) for the Reagents and Tools Table can be found in our author guidelines (section 'Structured Methods'):

14) Please order the manuscript sections like this, using only these names:

Title page - Abstract - Keywords - Introduction - Results - Discussion - Methods - Data availability section - Acknowledgements (please put here all the funding information) - Disclosure and Competing Interests Statement - References - Figure legends - Expanded View Figure legends

15) Please make sure that all the funding information is also entered into the online submission system and that it is complete and similar to the one in the acknowledgement section of the manuscript text file.

Please note that corresponding authors are required to supply an ORCID ID upon submission of a revised manuscript. Please find instructions on how to link the ORCID ID to the account in our manuscript tracking system in our Author guidelines:

<http://www.embopress.org/page/journal/14693178/authorguide#authorshipguidelines>

I look forward to seeing a revised form of your manuscript when it is ready.

Yours sincerely,

Referee #1:

The manuscript by Basak et al shows that S1P-S1PR1 signaling elevated in tumor-infiltrating CD8 T cells activates the PERK-CHOP axis of the endoplasmic reticulum stress response, which then increases Map3k transcription and p38MAPK activation. Activated p38MARK in turn impairs CD8 T cell metabolism and effector function, as well as increasing the T cell apoptosis. Blocking this axis improves anti-PD-1 therapy. Overall, the results support the conclusion and there are some novel aspects of the study. However, there are a few items that need to be clarified before publication:

Fig. 1I and K,

Fig 2, the legend says n=4, is the experiment repeated?

Fig. 4. What does n=3 mean for the Western blots. Three times Western, if so, please quantify. C and D the quality can be better

Fig. 5. Western blots, n=3, please quantify using all the blots

Missing figure labeling-except Figure S2. This makes it very difficult to read.

Referee #2:

The paper presented by Basak et. al discusses underexplored functions of S1PR1 signaling in T cells within the tumor microenvironment. Their findings implicate S1PR1 as a mediator of T cell dysfunction in the context of cancer and describe possible therapeutic targets to reinvigate T cell responses.

The data presented in this manuscript is of high quality and the manuscript is well written. Data support most of the conclusions made within the text, however, systemic treatments with drugs in mouse models could still have non-immune effects; though

mechanistic experiments to show lymphocyte-driven effects may be too onerous for revision as they would not significantly change conclusions of the study.

S1PR1 has implications in the formation of T resident memory cells (Skon, Nature Immunology, 2013), which should be discussed in the context of the author's findings as T resident memory cell phenotypes are connected with productive anti-tumor immunity across a spectrum of cancers (including melanoma). It is possible that the dysfunctional differentiation of T cells, aided by S1PR1, contributes to the terminal exhaustion effects seen in cancer.

Referee #3:

Shilpak Chatterjee and colleagues investigate immunometabolism and tumor immunology, with a particular emphasis on how T cell metabolic states influence antitumor efficacy and responsiveness to immune checkpoint blockade. In this study, the authors explore the role of sphingosine-1-phosphate (S1P) signaling in modulating CD8⁺ T cell function within the tumor microenvironment (TME). They report that S1PR1 is highly expressed in tumor-infiltrating T cells, and that tumor-derived S1P promotes T cell death. Through pharmacological, transcriptomic, and epigenetic analyses, the study identifies the S1PR1-p38 MAPK axis as a critical mediator of these effects. Mechanistically, S1PR1-PERK signaling is shown to induce CHOP expression, which subsequently upregulates Map3k13 and Map3k15, leading to the activation of p38 MAPK and driving T cells toward an exhausted state. Inhibition of the p38 pathway is reported to restore T cell function in preclinical models. While the study offers novel insights into how tumors may exploit S1P signaling to suppress immunity, additional mechanistic and translational evidence will be necessary to strengthen the proposed model.

Major Comments:

1. The manuscript presents a controversial finding that S1P-S1PR1 signaling promotes T cell apoptosis, which conflicts with extensive literature supporting its role in T cell survival, proliferation, and trafficking¹. This fundamental discrepancy requires extensive discussion, robust experimental validation, and a clear reconciliation with established immunological principles. Without adequately addressing this, the core premise and generalizability of the study are severely undermined.
2. The study places S1PR1 upstream of the PERK-CHOP axis; however, the mechanistic basis for S1PR1-mediated PERK activation remains speculative. As S1PR1 can activate multiple canonical pathways such as RAS-MAPK, ERK, and PI3K/Akt, direct evidence is needed to demonstrate how it specifically engages PERK. Parallel perturbation experiments, for example, combining pharmacologic or genetic inhibition of the RAS pathway with S1PR1 blockade, would help establish pathway specificity and exclude indirect activation through alternative effectors.
3. To strengthen the translational relevance, the authors should contextualize the S1PR1-PERK-CHOP-p38 MAPK axis in relation to S1PR1's established downstream pathways. In particular, a comparative discussion outlining the potential advantages, such as improved specificity, reduced off-target effects, and applicability in checkpoint blockade refractory tumors, together with the possible limitations of targeting this noncanonical pathway, would help clarify its novelty and clinical potential.
4. Despite identifying a novel, noncanonical S1PR1-PERK-CHOP axis, the authors ultimately rely on a conventional p38 MAPK inhibitor for their therapeutic models. This approach reduces both the practical value and the novelty of identifying the upstream PERK and CHOP components. To strengthen the translational relevance of these findings, it is important to demonstrate the efficacy of directly targeting PERK or CHOP, or to provide a clear rationale for why p38 inhibition is the preferred or only feasible therapeutic strategy within the context of the proposed pathway.
5. The evidence for PERK's role is primarily correlative. Definitive demonstration of causality will require robust genetic approaches such as PERK knockout or siRNA-mediated silencing within the tumor microenvironment, as well as complementary gain-of-function studies. Furthermore, because PERK is a central mediator of the unfolded protein response and can be activated by DNA damage, both of which can independently induce CHOP, it is important to delineate the contribution of these alternative inputs. Employing pathway-specific inhibitors or activators and assays for DNA damage, such as the comet assay, would help exclude confounding mechanisms and reinforce the proposed signaling hierarchy.

Minor Comments:

1. Image Quality: Several Western blot panels lack optimal resolution and contrast. Please provide higher-quality images to ensure accurate interpretation of protein expression data.
2. Figure Formatting: Ensure consistent font styles, sizes, and panel formatting across all figures for a cohesive and professional presentation.
3. Acronym Definition (PERK) - Expand PERK to "protein kinase R (PKR)-like endoplasmic reticulum kinase (EIF2AK3)" at first mention in the Introduction to aid readers unfamiliar with ER stress pathways.
4. Acronym Definition (CHOP) - Expand CHOP to "C/EBP homologous protein" upon first mention, noting that the acronym has alternative meanings in oncology (e.g., chemotherapy regimens).

Reference

1. Dixit D, Hallisey VM, Zhu EY, et al. S1PR1 inhibition induces proapoptotic signaling in T cells and limits humoral responses within lymph nodes. *J Clin Invest.* 2024;134(4):e174984. Published 2024 Jan 9. doi:10.1172/JCI174984

Referee #4:

The authors investigate if S1PR1-S1P axis contributes to CD8 T cell exhaustion. S1P is associated with a terminal dysfunctional program in tumor infiltrating CD8 T cells. Mechanistically, S1P axis upregulates the CHOP axis in the ER and MAPK signaling that promotes CD8 T cell exhaustion. Inhibiting the MAPK pathway shows synergy with anti-PD-1 in an adoptive T cell model. The following questions need to be addressed.

Major comments:

1. How viable is a systemic therapeutic targeting this axis since it contributes to migration of T cells from the lymph nodes to the tumor. What are the results of targeting this pathway on CD8 T cell egress from the lymph node.
2. Why did the authors use an adoptive T cell transfer model, what are the results in a wildtype mouse tumor model with anti-PD-1 and p38i combination therapy.
3. The authors need to include data to support that the in vitro method to produce terminal exhausted CD8 T cells did indeed generate exhausted CD8 T cells.

Minor comments:

1. Figure 3 legend: it is not clear if CD8 T cells are from the tumor or generated in vitro.
2. How were CD8 T cells activated in mouse in vitro chronic exhaustion experiment.
3. Legends for the bar plots are missing.
4. The dosing scheme of the anti-PD-1 antibody should be included in figure legends- what are the total doses administered. Anti-PD-1 antibody clone name should be included.
5. Method section and figure legends should include all details necessary to understand an experiment.

Response to reviewer's comments (MS# EMBOR-2025-62267V1)

We thank the reviewers for their helpful critiques and suggestions. We have taken the reviewer's concerns and suggestions into consideration and revised this submission accordingly. The key concerns raised by the specific reviewers have been addressed below:

Reviewer #1

1. Fig. 1I and K,

Response: We thank the reviewer for the comment and provide the following clarification:

Figure 1I shows representative flow cytometry dot plots along with quantification of intracellular effector molecules - Granzyme B (GZMB), IFN- γ , and TNF- α - in murine CD8⁺ T cells activated for three days in the presence or absence of S1P. As indicated in the figure legend, all experiments were independently repeated seven times (n=7).

Figure 1K presents representative Annexin V/7-AAD flow cytometry plots and corresponding bar graphs quantifying cell viability in murine CD8⁺ T cells activated for three days in the presence or absence of S1P, followed by overnight re-stimulation with anti-CD3. The experimental details are provided in the Materials and Methods section. As noted in the figure legend, the experiments were independently repeated five times (n=5).

2. Fig 2, the legend says n=4, is the experiment repeated?

Response: We thank the reviewer for their question. To clarify, this experiment was performed once, and $n = 4$ represents four biological replicates obtained from four individual mice.

3. Fig. 4. What does n=3 mean for the Western blots. Three times Western, if so, please quantify. C and D the quality can be better

Response: We thank the reviewer for pointing this out. To clarify, $n = 3$ indicates that the western blot data were generated from three independent biological replicates. As suggested, we have now included a bar graph adjacent to the representative blot showing cumulative quantification from the biological replicates.

4. Fig. 5. Western blots, n=3, please quantify using all the blots
Missing figure labeling-except Figure S2. This makes it very difficult to read.

Response: We thank the reviewer for the comment. As suggested, we have now included a bar graph adjacent to the representative blot, showing cumulative quantification from the biological replicates. Additionally, all figures have been properly labeled as requested.

Reviewer #2

We thank the reviewer for appreciating our work and acknowledging the novelty of our findings that identify S1PR1 as a mediator of T cell dysfunction in the context of cancer, and highlight possible therapeutic targets to reinvigorate T cell responses.

S1PR1 has implications in the formation of T resident memory cells (Skon, Nature Immunology, 2013), which should be discussed in the context of the author's findings as T resident memory cell phenotypes are connected with productive anti-tumor immunity across a spectrum of cancers (including melanoma). It is possible that the dysfunctional differentiation of T cells, aided by S1PR1, contributes to the terminal exhaustion effects seen in cancer.

Response: We thank the reviewer for this insightful comment. We agree that S1PR1 plays a critical role in regulating T-cell tissue residency while inversely influencing the formation and maintenance of tissue-resident memory T (T_{RM}) cells, as demonstrated by Skon et al., 2013 (Nat. Immunol.), Mackay et al., 2013 (Nat. Immunol.), and Zhu et al., 2013 (Nature). Although higher frequencies of T_{RM} cells in tumors correlate with improved clinical outcomes, the mechanisms underlying their contribution to tumor control remain incompletely defined. Based on our findings showing that S1P-S1PR1 signaling suppresses the functional attributes of effector $CD8^+$ T cells, we argue that the inherently low S1PR1 expression on T_{RM} cells may help them evade S1P-mediated immune suppression within the S1P-rich tumor microenvironment. However, it remains unclear whether, following antigen recognition in tumors, T_{RM} cells can differentiate into effector progeny that re-express KLF2 and S1PR1. If this transition occurs, these cells may then become susceptible to S1P-driven dysfunction and reduced proliferative competence, similar to peripherally recruited tumor-reactive $CD8^+$ T cells. Therefore, therapeutic strategies directed against S1PR1 or its downstream signaling may be beneficial in the context of tumors, not only to restore the effector function and proliferative capacity of tumor-reactive $CD8^+$ T cells, but at the same time help maintain the functional integrity of T_{RM} cells.

The section below is now included in the discussion section:

Beyond driving effector $CD8^+$ T cell dysfunction, S1PR1 also plays a pivotal role in inversely regulating the development of tissue-resident memory T (T_{RM}) cells (Skon et al., 2013; Mackay et al., 2013; Zhu et al., 2013). Because T_{RM} cells characteristically express low levels of S1PR1, our findings suggest that they may be inherently protected from S1P-mediated suppression in the S1P-rich tumor microenvironment. This reduced susceptibility could contribute to the favorable association between T_{RM} abundance and improved tumor control. However, it remains unclear whether T_{RM} cells, upon re-encounter with antigen, generate effector progeny that re-express KLF2 and S1PR1, and thus become vulnerable to S1P-driven dysfunction, similarly observed in peripherally recruited tumor-reactive $CD8^+$ T cells. These insights raise the possibility that therapeutic inhibition of S1PR1 or its downstream signaling could simultaneously reinvigorate exhausted effector $CD8^+$ T cells while preserving the functional integrity of T_{RM} cells, thereby enhancing anti-tumor immunity on multiple fronts.

Reviewer # 3

1. *The manuscript presents a controversial finding that S1P-S1PR1 signaling promotes T cell apoptosis, which conflicts with extensive literature supporting its role in T cell survival, proliferation, and trafficking. This fundamental discrepancy requires extensive discussion, robust experimental validation, and a clear reconciliation with established immunological principles. Without adequately addressing this, the core premise and generalizability of the study are severely undermined.*

Response: We thank the reviewer for highlighting this important dichotomy in the role of S1P–S1PR1 signaling in T-cell survival. We acknowledge the findings by Dixit et al. (PMID: 38194271), who reported that S1P-S1PR1 signaling promotes survival in naïve T cells by limiting JNK activation and maintaining a favorable balance of BCL-2 family proteins. In contrast, our study demonstrates that in activated CD8⁺ T cells, S1P-S1PR1 signaling increases susceptibility to apoptosis following TCR stimulation. We propose that this discrepancy arises from distinct intracellular signaling networks operating in naïve versus activated T cells, which differentially shape the downstream consequences of S1P-S1PR1 engagement. Moreover, variations in S1PR1 expression levels between these T-cell states may further contribute to the opposing survival outcomes observed. We have now incorporated this perspective into the Discussion section to clearly convey the mechanistic distinction between naïve and activated T cells.

The section below is now included in the discussion section:

It is also noteworthy that although no overt loss of viability was observed in activated CD8⁺ T cells treated with exogenous S1P, the S1P-S1PR1 axis rendered them more susceptible to apoptosis following subsequent TCR stimulation. This pro-apoptotic effect stands in sharp contrast to findings in naïve T cells, where S1P-S1PR1 signaling has been shown to promote cell survival by limiting JNK activation and maintaining a favorable balance of BCL-2 family proteins (Dixit et al, 2024). We propose that this discrepancy reflects fundamental differences in the intracellular signaling networks (signalomes) that regulate naïve versus activated T cells, resulting in distinct downstream consequences of S1P-S1PR1 engagement. Additionally, the differential expression levels of S1PR1 between naïve and activated T cells may further shape the signaling cascades induced by S1P, ultimately determining the opposing survival outcomes observed.

2. The study places S1PR1 upstream of the PERK-CHOP axis; however, the mechanistic basis for S1PR1-mediated PERK activation remains speculative. As S1PR1 can activate multiple canonical pathways such as RAS-MAPK, ERK, and PI3K/Akt, direct evidence is needed to demonstrate how it specifically engages PERK. Parallel perturbation experiments, for example, combining pharmacologic or genetic inhibition of the RAS pathway with S1PR1 blockade, would help establish pathway specificity and exclude indirect activation through alternative effectors.

Response: We thank the reviewer for raising this important mechanistic point. In line with the suggestion, we examined canonical signaling pathways downstream of S1PR1 to determine whether any of them mediate PERK activation. As shown below, supplementation of activated CD8⁺ T cells with S1P did not result in appreciable changes in Ras, Raf, pERK1/2, or pAKT levels, indicating that classical RAS-MAPK and PI3K-AKT pathways are unlikely to drive PERK activation in this context. Although not included in the original manuscript, we also observed that S1P treatment led to increased reactive oxygen species (ROS) accumulation, and inhibition of ROS with N-acetyl cysteine (NAC) markedly reduced both p-PERK and CHOP levels. These observations collectively point toward a noncanonical mechanism of PERK activation, rather than classical S1PR1 downstream signaling.

Given that our RNA-seq analysis showed enrichment of ER-stress signatures in S1P-treated CD8⁺ T cells, and CHOP expression was robustly induced in this setting, our primary objective in this study was to dissect how S1P-S1PR1-mediated CHOP activation impairs CD8⁺ T cell function. Furthermore, because S1PR1 expression was predominantly restricted to terminally exhausted intratumoral CD8⁺ T

cells co-expressing PD-1 and Tim-3, we focused on understanding whether this axis regulates dysfunction-associated traits in these cells, rather than fully resolving the upstream mechanism governing PERK activation. While we agree that identifying the precise signaling cascade leading to PERK activation would complete the upstream arm of this pathway, we believe this mechanistic detail falls beyond the current scope of the study.

Importantly, we have initiated a complementary follow-up project that reveals a potential proteotoxicity-driven mechanism of PERK activation downstream of S1P-S1PR1, which we believe may explain the elevated CHOP levels observed here. We hope the reviewer will appreciate that the current manuscript was designed to unravel the functional consequences of the S1P-S1PR1-CHOP axis in exhausted CD8⁺ T cells, and we trust that the reviewer will consider our rationale for not emphasizing the upstream mechanism of PERK activation in this study.

(A-D) Western blot analysis showing the expression of (A) Ras, (B) Raf, (C) p-AKT and (D) p-ERK in in-vitro activated CD8⁺ T cells either in the presence or absence of S1P. The adjacent bar graph depicts normalized densitometric data from three independent experiments (n=3). (E) Flow cytometry analysis depicting mean fluorescence intensity of cytosolic ROS using DCFDA from CD8⁺ T cells activated in presence or absence of S1P. The adjacent bar graph represents cumulative data from four independent experiments (n=4). (F) Western blot analysis showing expression of CHOP in control, S1P, and NAC pretreated S1P-treated CD8⁺ T cells. The adjacent bar graph depicts normalized densitometric data from three independent experiments (n=3). *p < 0.05; **p < 0.01; ***p < 0.005; ****p < 0.0001; ns, nonsignificant. p values are derived from unpaired two-tailed Student's t-test (A-E), one-way Anova (F).

3. To strengthen the translational relevance, the authors should contextualize the S1PR1-PERK-CHOP-p38 MAPK axis in relation to S1PR1's established downstream pathways. In particular, a comparative discussion outlining the potential advantages, such as improved specificity, reduced off-target effects, and applicability in checkpoint blockade refractory tumors, together with the possible limitations of targeting this noncanonical pathway, would help clarify its novelty and clinical potential.

Response: We thank the reviewer for this insightful comment. We agree that placing the S1PR1-PERK-CHOP-p38 MAPK axis in the context of canonical S1PR1 signaling pathways strengthens the translational relevance of our findings. While S1PR1 is traditionally known to signal through STAT3, PI3K-AKT, and ERK to support T-cell survival, proliferation, and trafficking, our data identify a distinct, noncanonical cascade engaged under the chronic S1P exposure characteristic of the tumor microenvironment (TME). We show that persistent S1PR1 activation in activated CD8⁺ T cells induces PERK-CHOP-p38 MAPK signaling, leading to diminished effector function, reduced proliferative fitness, and increased apoptosis upon TCR stimulation. Selectively targeting the downstream S1P-S1PR1 signaling axis could offer several clinical advantages, including improved specificity by sparing

beneficial S1PR1 functions and reduced systemic toxicity compared with global S1PR1 blockade. At the same time, we acknowledge important limitations, including the need to delineate molecular conditions that drive a shift from canonical to ER-stress-associated signaling and to evaluate broader immunological consequences of interfering with PERK-CHOP pathways. As suggested, we have now incorporated a comparative discussion in the revised manuscript to highlight both the novelty and therapeutic potential of selectively modulating this noncanonical S1PR1 axis.

The section below is now included in the discussion section:

..... our study identifies the S1P-S1PR1-CHOP-p38 MAPK axis as a previously unrecognized regulatory circuit that contributes to CD8⁺ T cell dysfunction and attrition in tumors. These findings uncover new mechanistic underpinnings of T cell exhaustion and reveal p38 MAPK as a promising therapeutic target to enhance CD8⁺ T cell survival and responsiveness to ICB therapies in the TME. While S1PR1 has been primarily associated with activation of signaling pathways such as STAT3, PI3K-AKT, and ERK that promote T-cell survival, proliferation, and trafficking, our data uncover a distinct, stress-associated S1PR1-PERK-CHOP-p38 MAPK axis that emerges under the chronic S1P exposure characteristic of the TME. Selectively targeting this maladaptive signaling state may enhance therapeutic specificity by sparing beneficial S1PR1 functions and reducing systemic toxicity associated with global receptor inhibition. Collectively, these insights highlight the novelty and translational potential of modulating S1PR1-driven stress signaling to reinvigorate anti-tumor CD8⁺ T cell immunity.

4. Despite identifying a novel, noncanonical S1PR1-PERK-CHOP axis, the authors ultimately rely on a conventional p38 MAPK inhibitor for their therapeutic models. This approach reduces both the practical value and the novelty of identifying the upstream PERK and CHOP components. To strengthen the translational relevance of these findings, it is important to demonstrate the efficacy of directly targeting PERK or CHOP, or to provide a clear rationale for why p38 inhibition is the preferred or only feasible therapeutic strategy within the context of the proposed pathway.

Response: We thank the reviewer for this important suggestion. In response, we have expanded our study to directly assess the therapeutic impact of targeting PERK and CHOP, going beyond p38 MAPK inhibition.

In the revised version of the manuscript, we first provide supporting evidence that directly targeting CHOP offers a viable therapeutic strategy. Previously, we showed that the pharmacological inhibition of CHOP using GSK significantly delayed tumor progression in a subcutaneous B16F10 melanoma model and improved the effector function of intratumoral CD8⁺ T cells. To reinforce the translational impact, we have now added new data demonstrating that CHOP inhibition synergizes with anti-PD-1 treatment, improving therapeutic responsiveness similarly to p38 MAPK inhibition.

Furthermore, as suggested, we directly tested the role of PERK in T cell dysfunction. We performed shRNA-mediated PERK knockdown in tumor antigen-specific Pmel T cells and adoptively transferred these cells into B16F10 melanoma-bearing mice. The revised manuscript now includes data showing that PERK-deficient Pmel T cells exhibit enhanced anti-tumor activity, and this benefit is further augmented when combined with anti-PD-1 therapy. These results demonstrate that PERK constitutes

a viable therapeutic target that modulates CHOP-mediated impairment of T cell function in the tumor microenvironment.

5. The evidence for PERK's role is primarily correlative. Definitive demonstration of causality will require robust genetic approaches such as PERK knockout or siRNA-mediated silencing within the tumor microenvironment, as well as complementary gain-of-function studies. Furthermore, because PERK is a central mediator of the unfolded protein response and can be activated by DNA damage, both of which can independently induce CHOP, it is important to delineate the contribution of these alternative inputs. Employing pathway-specific inhibitors or activators and assays for DNA damage, such as the comet assay, would help exclude confounding mechanisms and reinforce the proposed signaling hierarchy.

Response: We thank the reviewer for this insightful comment. As noted in our response to Comment #2, the upstream mechanism of PERK activation in dysfunctional CD8⁺ T cells remains to be fully elucidated, although our ongoing work increasingly supports proteostatic stress as the predominant driver. Because the primary goal of this study was to dissect how S1P-S1PR1 induces CHOP activation and impairs CD8⁺ T cell function, we focused on validating the downstream signaling consequences and evaluating whether targeting CHOP or its regulators could restore anti-tumor immunity. While we agree that defining additional inputs to PERK activation, such as DNA damage-associated pathways, would further refine the upstream hierarchy, this mechanistic dissection lies beyond the scope of the current work.

To directly address causality, we incorporated a genetic loss-of-function approach targeting PERK. Specifically, we performed adoptive cell transfer using tumor-specific Pmel T cells with shRNA-mediated PERK silencing (Pmel^{shPERK}) in the B16-F10 melanoma model. Unlike wild-type Pmel cells, which showed limited and non-durable responses with or without anti-PD-1 therapy, PERK-deficient Pmel^{shPERK} cells exhibited markedly enhanced anti-tumor activity that was further potentiated by anti-PD-1 treatment. These findings provide direct functional evidence that PERK-driven CHOP activation downstream of S1P-S1PR1 is a key determinant of CD8⁺ T cell dysfunction and therapeutic resistance, establishing causality and reinforcing the proposed signaling hierarchy.

Minor Comments:

1. Image Quality: Several Western blot panels lack optimal resolution and contrast. Please provide higher-quality images to ensure accurate interpretation of protein expression data.

Response: As suggested, we have replaced the Western blot panels with higher-resolution images and have also included cumulative densitometric analyses from multiple biological replicates.

2. Figure Formatting: Ensure consistent font styles, sizes, and panel formatting across all figures for a cohesive and professional presentation.

Response: We apologize for the inconsistencies in the initial figure formatting. All figures have now been revised to ensure uniform font styles, sizes, and panel formatting.

3. Acronym Definition (PERK) - Expand PERK to "protein kinase R (PKR)-like endoplasmic reticulum kinase (EIF2AK3)" at first mention in the Introduction to aid readers unfamiliar with ER stress pathways.

Response: We thank the reviewer for the suggestion. We have now expanded the acronym PERK to "protein kinase R (PKR)-like endoplasmic reticulum kinase (EIF2AK3)" at its first mention in the Introduction.

4. Acronym Definition (CHOP) - Expand CHOP to "C/EBP homologous protein" upon first mention, noting that the acronym has alternative meanings in oncology (e.g., chemotherapy regimens).

Response: We thank the reviewer for the suggestion. We have now expanded the acronym CHOP to "C/EBP homologous protein" at its first mention in the Introduction.

Reviewer # 4

Minor comments:

1. *Figure 3 legend: it is not clear if CD8 T cells are from the tumor or generated in vitro.*

Response: We are deeply sorry for the confusion. This is to clarify that CD8⁺ T cells were generated in vitro by activating them with anti-CD3 and anti-CD28 for 3 days, and divided into two groups: Control and S1P-treated cells.

2. *How were CD8 T cells activated in a mouse in vitro chronic exhaustion experiment?*

Response: For the mouse in vitro chronic exhaustion assay, CD8⁺ T cells were first activated for 48 hours using plate-bound anti-CD3 (5 µg/ml) and anti-CD28 (5 µg/ml) in the presence of IL-2 (100 U/ml). After initial activation, cells were subjected to continuous stimulation with plate-bound anti-CD3 (5 µg/ml) for an additional 6 days, with cultures maintained at optimal density by splitting every 2 days and replenishing fresh medium with IL-2. This prolonged stimulation protocol mimics persistent antigen exposure and induces an exhaustion phenotype, as previously described (REF). We have now included a comprehensive phenotypic characterization of these in vitro-exhausted CD8⁺ T cells in the Supplementary Figures.

3. *Legends for the bar plots are missing.*

Response: We apologize for this oversight. The missing legends for the bar plots have now been added in the revised manuscript.

4. *The dosing scheme of the anti-PD-1 antibody should be included in figure legends- what are the total doses administered. Anti-PD-1 antibody clone name should be included.*

Response: We thank the reviewer for the suggestion. The dosing scheme and antibody details have now been added to the figure legends.

5. *Method section and figure legends should include all details necessary to understand an experiment.*

Response: We agree with the reviewer's recommendation. The Methods section and figure legends have now been revised to include all essential experimental details.

Major Comments:

1. How viable is a systemic therapeutic targeting this axis since it contributes to the migration of T cells from the lymph nodes to the tumor. What are the results of targeting this pathway on CD8 T cell egress from the lymph node?

Response: We thank the reviewer for this important question. In our initial attempts, YUMM1.7 melanoma-bearing mice were treated systemically with an S1PR1 inhibitor via intraperitoneal injection. This approach was not viable, as systemic blockade of S1PR1 impairs T cell egress and homeostasis, resulting in marked lymphodepletion and poor survival. To avoid these adverse effects, we therefore shifted to intratumoral delivery, allowing localized inhibition of S1PR1 within the tumor microenvironment while sparing systemic lymphocyte trafficking.

Furthermore, recognizing the limitations of directly targeting S1PR1, we focused on identifying and modulating the downstream signaling mediators responsible for CD8⁺ T cell dysfunction. Our data show that selectively targeting PERK, CHOP, or p38 MAPK downstream of S1P-S1PR1 signaling effectively reverses CD8⁺ T cell dysfunction and enhances anti-PD-1 responsiveness without interfering with T cell egress. This strategy therefore provides a more feasible and translationally relevant therapeutic approach.

2. Why did the authors use an adoptive T cell transfer model, what are the results in a wildtype mouse tumor model with anti-PD-1 and p38i combination therapy.

Response: We used the adoptive T cell transfer model to specifically evaluate the role of p38 MAPK signaling in tumor antigen-specific CD8⁺ T cells. The B16-F10 model naturally expresses the gp100 antigen, enabling the use of Pmel-1 transgenic CD8⁺ T cells that recognize gp100 in an MHC class I-restricted manner. This approach allowed us to directly assess how p38 MAPK inhibition influences the phenotype, effector function, and anti-PD-1 responsiveness of tumor-specific CD8⁺ T cells within the TME.

In response to the reviewer's comment, we further evaluated the combination of anti-PD-1 and p38 MAPK inhibition in a wild-type mouse tumor model without adoptive transfer. Consistent with our ACT-based observations, the combined therapy led to delayed tumor progression, improved CD8⁺ T cell effector function, and reduced exhaustion, demonstrating that the therapeutic benefit of p38 MAPK inhibition extends to standard endogenous T cell responses.

3. The authors need to include data to support that the in vitro method to produce terminal exhausted CD8 T cells did indeed generate exhausted CD8 T cells.

Response: We thank the reviewer for the comment. We have now included extensive phenotypic and functional characterization of both human and murine chronically stimulated CD8⁺ T cells, demonstrating that our in vitro protocol successfully generates terminally exhausted CD8⁺ T cells. The figures are included as Extended View Figures (EV1D to EV1M).

Summary of response to reviewers:

We sincerely thank the reviewers for their constructive feedback. The comments significantly helped us strengthen our findings and further establish the S1P-S1PR1 axis as a critical immune checkpoint governing CD8⁺ T cell function and responsiveness to anti-PD-1 therapy. We believe that the revisions and additional experiments incorporated in the manuscript effectively address the points raised and enhance both the rigor and impact of our study. We hope that the revised submission will meet the reviewers' approval for publication.

Dear Dr. Chatterjee,

Thank you for the submission of your revised manuscript to our editorial offices. I have now received the reports from the two referees that I asked to re-evaluate the study, you will find below. As you will see, the referees now fully support publication of your study in EMBO reports. Nevertheless, referee #4 has two remaining concerns or suggestions to improve the manuscript, I ask you to address in a final revised manuscript. Please also provide a final p-b-p-response regarding the remaining referee points and the editorial requests below.

Editorial requests:

- Please provide a more active title. How about:

S1PR1-S1P signaling impairs CD8⁺ T cell metabolism and effector function in tumors

- Please provide individual production quality figure files as .eps, .tif, .jpg (one file per figure), of main figures and EV figures. We can't proceed with files in .pptx format. Moreover, please note that supplementary figures should be named 'Expanded View Figures' (in the legend and their file name), not 'Extended View Figures'. The nomenclature 'Figure EVx' is correct.

- We now use CRediT to specify the contributions of each author in the journal submission system. CRediT replaces the author contribution section. Please use the free text box to provide more detailed descriptions and do NOT provide your final manuscript text file with an author contributions section. See also our guide to authors (section 'Author contributions'): <https://link.springer.com/journal/44319/submission-guidelines#cms-Revised-submissions>

- Please remove now the referee access information from the Data Availability section and make sure that the dataset is public latest upon online publication of the manuscript. Please also add a direct link to the dataset. Please remove all other text regarding information availability from this section. Here only information regarding externally deposited datasets should be provided.

- Please check again that the number "n" for how many independent experiments were performed, their nature (biological versus technical replicates), the bars and error bars (e.g. SEM, SD) and the test used to calculate p-values is indicated in the respective figure legends (main, EV and Appendix figures). Please also check that all the p-values are explained in the legend, and that these fit to those shown in the figure. Please provide statistical testing where applicable. Please avoid the phrase 'independent experiment' but clearly state if these were biological or technical replicates. Please also indicate (e.g. with n.s.) if testing was performed, but the differences are not significant. In case n=2, please show the data as separate datapoints without error bars and statistics. See also:

<https://link.springer.com/journal/44319/submission-guidelines#cms-Figure-and-data-presentation>

If n<5, please show single datapoints for diagrams. Moreover:

- Please note that the exact p values are not provided in the legends of figures 1A-K, M, N, Q, R, S; 2A-E, G-J; 4A, B, C, D, E, F, H, I, J, K, L; 5A-F, G-I; 6B, C, D, F, G, H, J, K, L; 7B-E; EV1 A-C, D-G, I-L, N-Q; EV2 A, D; EV3 B, EV5 A-C

- Please indicate the statistical test used for data analysis in the legend of figure 3B

- Please note that information related to n is missing in the legends of figures 2C-E ; 6F, G, H, J-L

- Please note that the error bars are not defined in the legends of figures 1A-N, P-S; 2A, B, C, D, E, G, H, I, J; 4A, B, C, D, E, F, H, I, J, K, L; 5A-F, G-J; 6B, C, D, F, G, H, J, K, L; 7B-E; EV1 A-C, D-G, I-L, N-Q; EV2 A, D; EV3 B, EV4 A, B; EV5 A-C

- Please make sure that all the funding information is also entered into the online submission system and that it is complete and similar to the one in the acknowledgement section of the manuscript text file. Presently, the grants MLP-121 and P50 from CSIR-IICB, Kolkata, the Central Instrumentation Facility (CIF) of CSIR-IICB, Kolkata; the High Performance and Cloud Computing Group at the Zentrum für Datenverarbeitung of the University of Tübingen, the state of Baden-Württemberg through bwHPC, and the German Research Foundation (DFG) through grant no. INST 37/935-1 FUGG. de.NBI Cloud within the German Network for Bioinformatics Infrastructure (de.NBI) and ELIXIR-DE (Forschungszentrum Jülich and W-de.NBI-001, W-de.NBI-004, W-de.NBI-008, W-de.NBI-010, W-de.NBI-013, W-de.NBI-014, W-de.NBI-016, W-de.NBI-022) are missing in the online system. Please check.

- All Materials and Methods need to be described in the main text using our 'Structured Methods' format, which is required for all research articles. According to this format, the Methods section should include a Reagents and Tools Table (listing key reagents, experimental models, software, and relevant equipment and including their sources and relevant identifiers), uploaded as separate file, and a Methods section in which we encourage the authors to describe their methods using a step-by-step protocol format with bullet points, to facilitate the adoption of the methodologies across labs. More information on how to adhere to this format as well as downloadable templates (.doc) for the Reagents and Tools Table can be found in our author guidelines (section 'Structured Methods'):

<https://link.springer.com/journal/44319/submission-guidelines#cms-Manuscript-organisation-and-formatting>

Please include all the primer, antibody and reagents information (presently Tables 1, 2 and 3) into the Reagents & Tools table. Then please remove Tables 1-3 from the manuscript text file. Please add callouts to the R&T table to the methods section where appropriate.

- Please add scale bars of similar style and thickness to all microscopic images, using clearly visible black or white bars (depending on the background). Please place these in the lower right corner of the images themselves. Please do not write on or near the bars in the image but define the size in the respective figure legend. Thus, please provide scale bars without text for panel EV2B (and define the bar in the respective legend).

- Please provide the numerical source data as excel files.

In addition, I would need from you uploaded separately:

- a schematic summary figure as separate file that provides a sketch of the major findings (not a data image) in jpeg or tiff format (with the exact width of 550 pixels and a height of not more than 400 pixels) that can be used as a visual synopsis on our website. The image provided contains partly text that is not readable. Please provide this with bigger fonts.

I look forward to seeing the further revised version of your manuscript when it is ready. Please let me know if you have questions regarding the revision.

Best,

Referee #3:

The authors have addressed my questions.

Referee #4:

The authors Basak et al. have made the requested changes. A few points still need to be addressed,

1. Experiments that have been conducted once should be conducted at least twice for scientific rigor and reproducibility. This includes Figure 2.
2. Methods section should include more details; the authors should state the program used for electroporation, provide details including cell numbers used for in-vitro assays including lentiviral transduction or in-vitro T cell exhaustion, the sequences of shRNA and siRNA, the lentiviral vector backbone to name a few.

Response to reviewer's comments (MS# EMBOR-2025-62267V2)

We thank the reviewer for their constructive critiques and suggestions. We have carefully considered the comments and revised the manuscript accordingly. The key concerns raised by Reviewer #4 and the editorial office are addressed below:

Reviewer # 4

1. *Experiments that have been conducted once should be conducted at least twice for scientific rigor and reproducibility. This includes Figure 2.*

Response: During the earlier revision, following Reviewer# 1's query, we independently repeated all in vivo experiments and obtained highly consistent results. We did not include these additional data in the figures because, in each experiment, four independent mice per group, with each mouse representing a true biological replicate, already constitute the appropriate and statistically valid unit of replication for in vivo studies. Statistical analyses were therefore conducted across independent animals rather than technical repeats. This design meets accepted standards for rigor and reproducibility in animal-based immunology studies, and additional cohorts would not increase biological independence beyond this.

2. *Methods section should include more details; the authors should state the program used for electroporation, provide details including cell numbers used for in-vitro assays including lentiviral transduction or in-vitro T cell exhaustion, the sequences of shRNA and siRNA, the lentiviral vector backbone to name a few.*

Response: As suggested, the Methods section has now been substantially expanded to include the electroporation program parameters, detailed cell numbers for all in vitro assays (including lentiviral transduction and in vitro T cell exhaustion experiments), the detailed information of shRNA and siRNA reagents, and the lentiviral vector backbone information.

Editorial requests

1. *Please provide a more active title. How about:*

"S1PR1-S1P signaling impairs CD8⁺ T cell metabolism and effector function in tumors."

Response: We agree with the editorial suggestion and have revised the manuscript title accordingly.

2. *Please provide individual production quality figure files as .eps, .tif, .jpg (one file per figure), of main figures and EV figures. We can't proceed with files in .pptx format. Moreover, please note that supplementary figures should be named 'Expanded View Figures' (in the legend and their file name), not 'Extended View Figures'. The nomenclature 'Figure EVx' is correct.*

Response: The production-quality .tif files for both the main figures and the EV figures have now been uploaded. The supplementary figures have now been named 'Expanded View Figures'.

3. *We now use CRediT to specify the contributions of each author in the journal submission system. CRediT replaces the author contribution section. Please use the free text box to provide more detailed descriptions and do NOT provide your final manuscript text file with an author contributions section.*

Response: The author contribution section has been removed from the manuscript text file, and all author contributions have been specified using CRediT in the journal submission system.

4. Please remove now the referee access information from the Data Availability section and make sure that the dataset is public latest upon online publication of the manuscript. Please also add a direct link to the dataset. Please remove all other text regarding information availability from this section. Here only information regarding externally deposited datasets should be provided.

Response: The referee access information has been removed from the Data Availability section. The data are publicly available under the accession number GSE313941.

5. Please check again that the number "n" for how many independent experiments were performed, their nature (biological versus technical replicates), the bars and error bars (e.g. SEM, SD) and the test used to calculate p-values is indicated in the respective figure legends (main, EV and Appendix figures). Please also check that all the p-values are explained in the legend, and that these fit to those shown in the figure. Please provide statistical testing where applicable. Please avoid the phrase 'independent experiment' but clearly state if these were biological or technical replicates. Please also indicate (e.g. with n.s.) if testing was performed, but the differences are not significant. In case n=2, please show the data as separate datapoints without error bars and statistics.

Response: We have carefully revised all figure legends (main and EV) to clearly report the number of replicates (n), specify whether they represent biological or technical replicates, define error bars (SEM or SD), and state the statistical tests used. All p-values are now fully explained and verified to match those shown in the figures, and statistical testing has been provided wherever applicable. Non-significant comparisons are indicated (e.g., ns).

5. Please note that the exact p values are not provided in the legends of figures 1A-K, M, N, Q, R, S; 2A-E, G-J; 4A, B, C, D, E, F, H, I, J, K, L; 5A-F, G-I; 6B, C, D, F, G, H, J, K, L; 7B-E; EV1 A-C, D-G, I-L, N-Q; EV2 A, D; EV3 B, EV5 A-C

Response: To avoid confusion in defining all panels in individual figures, exact p-values are provided in the corresponding source data excel files, while p-value summaries are included in the figure legends.

6. Please indicate the statistical test used for data analysis in the legend of figure 3B

Response: The statistical test used for data analysis in Figure 3B has now been included in the figure legend.

7. Please note that information related to n is missing in the legends of figures 2C-E ; 6F, G, H, J-L

Response: The missing information on n has now been added to the legends of Figures 2C-E and 6F-H, J-L.

8. Please note that the error bars are not defined in the legends of figures 1A-N, P-S; 2A, B, C, D, E, G, H, I, J; 4A, B, C, D, E, F, H, I, J, K, L; 5A-F, G-J; 6B, C, D, F, G, H, J, K, L; 7B-E; EV1 A-C, D-G, I-L, N-Q; EV2 A, D; EV3 B, EV4 A, B; EV5 A-C

Response: The error bars are now clearly defined in the legends for all the indicated figures.

9. Please make sure that all the funding information is also entered into the online submission system and that it is complete and similar to the one in the acknowledgement section of the manuscript text file. Presently, the grants MLP-121 and P50 from CSIR-IICB, Kolkata, the Central Instrumentation Facility (CIF) of CSIR-IICB, Kolkata; the High Performance and Cloud Computing Group at the Zentrum für Datenverarbeitung of the University of Tübingen, the state of Baden-Württemberg through bwHPC, and the German Research Foundation (DFG) through grant no. INST 37/935-1 FUGG. de.NBI Cloud within the German Network for Bioinformatics Infrastructure (de.NBI) and ELIXIR-DE (Forschungszentrum Jülich and W-de.NBI-001, W-de.NBI-004, W-de.NBI-008, W-de.NBI-010, W-de.NBI-013, W-de.NBI-014, W-de.NBI-016, W-de.NBI-022) are missing in the online system. Please check.

Response: All funding information has now been entered into the online submission system.

10. All Materials and Methods need to be described in the main text using our 'Structured Methods' format, which is required for all research articles. According to this format, the Methods section should include a Reagents and Tools Table (listing key reagents, experimental models, software, and relevant equipment and including their sources and relevant identifiers), uploaded as separate file, and a Methods section in which we encourage the authors to describe their methods using a step-by-step protocol format with bullet points, to facilitate the adoption of the methodologies across labs.

Response: The Methods section has been fully revised to comply with the journal's Structured Methods format. A Reagents and Tools Table has been prepared and included in the manuscript as well as uploaded as a separate file, and the Methods section has been rewritten in a step-by-step, bullet-point protocol format.

11. Please include all the primer, antibody and reagents information (presently Tables 1, 2 and 3) into the Reagents & Tools table. Then please remove Tables 1-3 from the manuscript text file. Please add callouts to the R&T table to the methods section where appropriate.

Response: All the reagents, primers, antibodies, and other relevant information have now been included in the Reagents & Tools table.

12. Please add scale bars of similar style and thickness to all microscopic images, using clearly visible black or white bars (depending on the background). Please place these in the lower right corner of the images themselves. Please do not write on or near the bars in the image but define the size in the respective figure legend. Thus, please provide scale bars without text for panel EV2B (and define the bar in the respective legend).

Response: Scale bars of uniform style and thickness have now been added to all microscopic images and placed in the lower right corner. Text has been removed from the images, and scale bar sizes are defined in the corresponding figure legends, including for panel EV2B.

13. Please provide the numerical source data as excel files.

Response: The numerical source data have now been provided as excel files.

14. A schematic summary figure as separate file that provides a sketch of the major findings (not a data image) in jpeg or tiff format (with the exact width of 550 pixels and a height of not more than 400 pixels) that can be used as a visual synopsis on our website. The image provided contains partly text that is not readable. Please provide this with bigger fonts.

Response: A schematic summary figure of high resolution has been uploaded.

Shilpak Chatterjee
CSIR-Indian Institute of Chemical Biology
Cancer Biology and Inflammatory Disorder
India

Dear Dr. Chatterjee,

Thank you for the submission of your final revised manuscript to our editorial offices. It now went through this and your final p-b-p-response and consider the remaining points of referee #4 and the editorial requests as adequately addressed.

I am thus very pleased to accept your manuscript for publication in the next available issue of EMBO reports. Thank you for your contribution to our journal.

You may qualify for financial assistance for your publication charges - either via a Springer Nature fully open access agreement or an EMBO initiative. Check your eligibility: <https://link.springer.com/journal/44319/how-to-publish-with-us>

Yours sincerely,

>>> Please note that it is EMBO Reports policy for the transcript of the editorial process (containing referee reports and your response letter) to be published as an online supplement to each paper. If you do NOT want this, you will need to inform the Editorial Office via email immediately. More information is available here: <https://link.springer.com/partners/embo-press/editorial-policies#Peer%20review>